# The Fully Nonlinear Stratified Geostrophic Adjustment Problem

Aaron Coutino[1] and Marek Stastna[1]

[1]University of Waterloo, Department of Applied Mathematics

*Correspondence to:* Aaron Coutino (acoutino@uwaterloo.ca)

**Abstract.** The study of the adjustment to equilibrium by a stratified fluid in a rotating reference frame is a classical problem in geophysical fluid dynamics. We consider the fully nonlinear, stratified adjustment problem from a numerical point of view. We present results of smoothed dam break simulations based on experiments in the published literature, with a focus on both the wave trains that propagate away from the nascent geostrophic state and the geostrophic state itself. We demonstrate that for Rossby numbers in excess of roughly 2 the wave train cannot be interpreted in terms of linear theory. This wave train consists of a leading solitary-like packet, and a trailing tail of dispersive waves. However, it is found that the leading wave packet never completely separates from the trailing tail. Somewhat surprisingly, the inertial oscillations associated with the geostrophic state exhibit evidence of nonlinearity even when the Rossby number falls below 1. We vary the width of the initial disturbance and the rotation rate so as to keep the Rossby number fixed, and find that while the qualitative response remains consistent, the Froude number varies and these variations are manifested in the form of the emanating wave train. For wider initial disturbances we find clear evidence of a wave train that initially propagates toward the near wall, reflects and propagates away from the geostrophic state behind the leading wave train. We compare kinetic energy inside and outside of the geostrophic state, finding that for long times a Rossby number of around one quarter yields an equal split between the two, with lower (higher) Rossby numbers yielding more energy in the geostrophic state (wave train). Finally we compare the energetics of the geostrophic state as the Rossby number varies finding long lived inertial oscillations in the majority of cases, and a general agreement with past literature that employed either hydrostatic, shallow water equation-based theory or stratified Navier-Stokes equations with a linear stratification.

## 1 Introduction

Geostrophic balance, namely the balance between the pressure gradient and the Coriolis pseudo force, is observed to hold to a good approximation for many large-scale motions in the atmosphere and the ocean. The process through which some disturbed state reaches this balance is called geostrophic adjustment. The linear problem was first considered by Rossby (1937). Using conservation of momentum and mass, he derived the geostrophic steady state corresponding to an initial perturbation. In the original publication, Rossby noted that the final state of the system possessed less energy than the initial state. The cause of this difference was identified by Cahn (1945), who showed that the end state is reached via inertial oscillations, which disperse energy through waves. Since then, numerous papers have used a variety of methods such as asymptotic expansions and numerical integration to solve this linear problem. There has been a great deal of published work on the linear problem

(Ou (1984); Gill (1976); Middleton (1987); Washington (1964); Mihaljan (1963)), but little on the fully nonlinear one. This is partly because nonlinear problems rarely yield analytical solutions in closed form, and partly because numerical methods applied to the problem must accurately resolve multiple length scales.

Kuo and Polvani (1999), a key paper in the study of the nonlinear problem, considered the adjustment problem in the context of the single layer shallow water equations in one dimension. The authors built on the results of Killworth (1992), and performed a numerical analysis of the fully nonlinear problem with 'dam break' initial conditions (see Gill (1982)). The authors found that the nonlinearity and rotation led to bore generation, with the bores dissipating energy as they propagated away from the geostrophic state. Since the nonlinear shallow water equations neglect non-hydrostatic dispersion, these bores manifested as shock-like fronts. This is in contrast to the non rotating stratified adjustment problem which leads to the generation of either a rank-ordered train of internal solitary waves, or an undular bore. Indeed, in this dispersive system, for the majority of parameter space, breaking is not observed. The authors also found that the inertial oscillations within the geostrophic state can persist for long times and are highly dependent on the initial conditions. In their analysis of the energy within the geostrophic state, the authors found that the ratio of change in kinetic energy ($\Delta$KE) to change in potential energy ($\Delta$PE) tended to $\frac{1}{3}$ which is the theoretically predicted value for both the linear and nonlinear problems (see Boss and Thompson (1995)). However, Kuo and Polvani showed that even for late times, the energy ratio fluctuated by up to 30% around the $\frac{1}{3}$ value.

While primarily viewed from a theoretical framework, rotation-modified adjustment has been shown to arise naturally in the ocean. One example of this are upwelling fronts which can create the initial density anomaly that then must adjust, see Chia et al. (1982) for a more complete discussion. More recently, Ledwell et al. (2004), and Oakey and Greenan (2004), as part of the Coastal Mixing and Optics experiment, showed the presence of patches of well mixed regions (density anomalies) throughout a background of stable fluid along the New England shelf. The specifics of this adjustment were investigated in Lelong and Sundermeyer (2005). The authors performed fully 3D numerical simulations of moderate resolution, using the nonhydrostatic equations under the Boussinesq approximation, of the adjustment process resulting from one of these density anomaly patches. To allow for more reasonable computation, the authors followed the procedure outlined in Lelong and Dunkerton (1998), and reduced the physical ratio for $\frac{N}{f}$ (by varying $f$ and holding $N$ constant), where $N$ is the buoyancy frequency and $f$ is the Coriolis parameter. For analysis, they separated the energy into kinetic and potential, and separated the domain into two regions, an inner region associated with the geostrophic state and an outer region associated with the waves. Since the initial conditions are static, the initial energy of the system is contained solely as potential energy. By comparing the energy within different areas of the simulation to the initial energy, the authors found that initial conditions with $\frac{Ro_r}{L} = 1$, where $Ro_r$ is the Rossby radius of deformation, and $L$ is the half-width of the initial state, were the most effective at generating kinetic energy in the geostrophic state. This is in contrast to cases with $\frac{Ro_r}{L} < 1$, where rotation effects dominate and little potential energy is converted to kinetic, or cases with $\frac{Ro_r}{L} > 1$, where the potential energy is primarily converted to wave energy. In all cases considered by these authors the radiated wave train was weak, composed of long waves and well approximated by linear theory. The nonlinear effects on the rotating adjustment problem have been investigated analytically using multiple scale perturbation analysis of the shallow-water and full stratified equations. In part one of a two part paper series Zeitlin et al. (2003a) perturb the rotating shallow-water equations using the Rossby number as their small parameter. The authors proceed to confirm that

a slow-fast splitting is possible, with the slow state largely remaining in geostrophic balance and largely unaffected by the fast state. In the waves that are generated, Zeitlin et al. observe shock formation and present a semi-quantitative criteria for this, based on the initial conditions. In the second paper, Zeitlin et al. (2003b), the authors generalize their results to the case of continuous stratification and also consider two layer and quasi-two layer stratifications. Zeitlin et al. perform a number of asymptotic expansions for different initial isopycnal deviation regimes. They conclude that for large deviations the model strongly depends on the ratio of the layer depths and that the waves produced from the initialization obey a Schrödinger-type modulation equation. For small deviations the waves generated are not impacted by the geostrophic state which is left to evolve according to the standard quasi-geostrophic (QG) equations.

Rotation-influenced nonlinear waves have also been considered using a model nonlinear wave equation; in this case a member of the Korteweg de Vries (KdV) family of equations . The KdV equation is the simplest model equation that allows for a balance between nonlinear and dispersive effects, with a rich mathematical structure which makes predictions of the evolution of an initial state that are remarkably robust in both a laboratory and field setting (see Johnson (1997)). A rotation-modified version of the KdV equation was first derived by Ostrovsky (see Grimshaw et al. (2012) for an in-depth discussion of the equation properties and references to the Russian literature). This new equation was subsequently analysed both through theoretical solutions found by asymptotic expansions and through numerical solutions. Investigation of the model equations revealed that the precise balance between nonlinearity and dispersion that leads to the traditional soliton solution of the KdV-equation is destroyed by the addition of rotation, and that over time the soliton breaks down into a nonlinear wave packet (Grimshaw and Helfrich (2008)). This hypothesis was later supported by experimental results, (Grimshaw et al. (2013)). From a theoretical point of view, Grimshaw and Helfrich (2008) also found that the extended nonlinear Schrödinger (NLS) equation provides a good qualitative description of the wave packet. While the mathematical developments of the rotation-modified theory are substantial, it is also true that this theory has a number of pathologies not observed in the non-rotating KdV-based theory, which in itself has been shown to misrepresent aspects of large amplitude solitary waves (Lamb (1997) is one of many papers to discuss some of these discrepancies).

Work has also been performed using models with higher order nonlinearity (Helfrich (2007)) but weak nonhydrostatic effects, as well as with the full set of stratified Euler equations (Stastna et al. (2009)). Both of these studies considered the breakdown of an initial solitary wave in the presence of rotation. Helfrich suggested that the initial solitary wave breaks down into a coherent leading wave packet with a trailing tail of waves. Stastna et al. suggested that for large amplitude, exact internal solitary waves that are solutions to the Dubreil-Jacotin-Long (DJL) equation, Helfrich's result was observed for artificially high rotation rates, while rotation rates typical of mid-latitudes led to a disturbance that never fully separated from the trailing tail. Despite differences in details, the qualitative features observed in both studies were quite similar. Additionally, they also performed collision experiments, finding that the packets that emerge from the initial solitary waves can merge during collisions, and hence do not interact as classical solitons. Finally, Stastna et al. also found that by increasing the width of a flat-crested wave, more energy was deposited into the tail. It remains to reconcile the two sets of results in detail, likely by systematically reducing the solitary wave amplitude used as an initial condition.

In this paper, we present the results of high resolution simulations of the geostrophic adjustment of a stratified fluid with a single

pycnocline on an experimental scale. Our simulations consider the full set of stratified Euler equations using a pseudo-spectral collocation method. We begin by providing and reviewing the non-rotating case and the changes that arise when polarity of the initial condition is changed. Next we present the general evolution of the rotating case using classical theory and two 'base' cases, one of which is comparable to one of the cases presented in Grimshaw et al. (2013). We subsequently identify the manner in which nonlinearity is exhibited in the problem, focusing on both the wave train and the geostrophic state and its inertial oscillations. We are able to clearly show the generation of a leftward propagating wave from the initial condition (especially evident for wider initial perturbations), and its subsequent reflection from the left wall. This wave train interacts with the geostrophic state, before and after reflecting off the left wall of the domain and then continues to propagate rightward across the tank. This is of potential interest to future experiments. We then focus on the geostrophic state in detail, specifically examining the change in kinetic energy and the change in potential energy for different initial widths, as well as the changes in the kinetic energy in the geostrophic state and the propagating wave train as the Rossby number varies. These results make the closest contact with the work of Lelong and Sundermeyer (2005). Finally we draw a number of conclusions based on our findings and identify directions for future work.

## 2  Methods

For the following numerical simulations, the full set of stratified Navier-Stokes equations for an incompressible fluid were used, though no span-wise variations were considered. Rotation was incorporated using an f-plane approximation and the non-traditional Coriolis terms were dropped. For a review of the effects of the non-traditional Coriolis terms see Gerkema et al. (2008). The x-axis is taken as parallel to the flat bottom with the z-axis pointing upward ($\hat{k}$ is the upward directed unit vector). The origin is placed in the bottom left corner so that both axes are positive. The incompressible Navier-Stokes equations for velocity $\boldsymbol{u} = [u(x,z,t), v(x,z,t), w(x,z,t)]$, density $\rho(x,z,t)$, and pressure $P(x,z,t)$ are,

$$\frac{D\boldsymbol{u}}{Dt} + (-fv, fu, 0) = -\boldsymbol{\nabla}P - \rho' g\hat{k} + \nu\nabla^2\boldsymbol{u}, \tag{1}$$

$$\nabla \cdot \boldsymbol{u} = 0, \tag{2}$$

$$\frac{D\rho}{Dt} = 0, \tag{3}$$

where $f$ is the constant Coriolis parameter, $g$ is acceleration due to gravity and $\nu$ is the kinematic viscosity. In accordance with convention, we have divided the momentum equation by the constant reference density $\rho_0$ and absorbed the hydrostatic pressure into the pressure $P$. We make the Boussinesq approximation for density and write $\rho = \rho_0(1 + \rho'(x,z,t))$, where $\rho'$ is considered a small perturbation. Due to our interest in the wave dynamics in the main water column, as opposed to details of the boundary layer dynamics, we impose free slip boundary conditions at the top and bottom of our domain. This will also ensure that the boundary layer does not play a significant role in the simulations on which we report. The walls allow us to mimic a lock-release set up that is used to create waves in many laboratory set ups (Carr and Davies (2006); Grue et al. (2000); Helfrich and Melville (2006)). We have chosen to neglect the span-wise dimension ($y$), as the lab results in Grimshaw et al. (2013) were

performed away from any side boundaries and the authors elected to neglect any curvature from the waves created. Another change from Grimshaw et al. (2013) is that we have a rigid lid as opposed to their free surface, this is due to the computational difficulty of a moving boundary.

In the following set of experiments the dominant dimensionless number is the Rossby number. This number is defined as
$Ro = \frac{U}{fL}$, where $U$ is the typical wave speed, $L$ is the typical length scale and $f$ is the Coriolis parameter. This reflects a ratio of the inertia term to the Coriolis pseudoforce term (henceforth just force). When the Coriolis force dominates, the fluid can reach a balance between the rotation and pressure terms, i.e., geostrophic balance. Since the fully equations contain the diffusion term $\nu\nabla^2$, it can be used to form the dimensionless Reynolds number which is given by $Re = \frac{UL}{\nu}$. $U$ and $L$ are the same as for $Ro$ and $\nu$ is the kinematic viscosity. The other relevant number considered is the Froude number $Fr = \frac{U}{c}$ which compares the typical wave speed $U$, to the theoretical wave speed $c$. In addition to these traditional dimensionless numbers we also define a nonlinearity parameter $\alpha$. Following from Kuo and Polvani (1997) this is defined as $\alpha = \frac{\eta}{H_1}$, where $\eta$ is the height of the displacement in isopycnals and $H_1$ is the height of the undisturbed fluid interface. This parameter can be used to modify the strength of the nonlinearity, and is well suited for shallow water equations. However, for the full set of incompressible Navier-Stokes equations some ambiguity is introduced by the vertical structure of the stratification and the initial perturbation. Nevertheless, we have found $\alpha$ to be a useful parameter, likely since the disturbances in our simulations are dominated by mode-1 waves.

The numerical simulations presented here were performed using an incompressible Navier Stokes equation solver which implements a pseudo-spectral collocation method (SPINS), presented in Subich et al. (2013). The solver uses spectral methods resulting in the order of accuracy scaling with the number of grid points. To deal with the build up of energy in the high wave numbers an exponential filter is used after a specific wave number cut off.

We computed a series of 2D lab scale numerical simulations on a similar scale to the physical experiments presented in Grimshaw et al. (2013), which were performed using the 13m diameter rotating platform at the LEGI-Coriolis Laboratory in Grenoble. Motivated by the results presented in Stastna et al. (2009), a domain four times longer than the physical tank ($Lx = 52$m) was used, as the 13m is an insufficient length when considering lower (closer to physical) rotation rates. In addition to this it was decided to change the tank depth to a more evenly divisible 0.4m (from a laboratory value of 0.36m). The density difference was set to 1% to match Grimshaw and Helfrich. The different physical parameters related to the initial set-up are illustrated in Fig. 1.

8192 grid points were used to resolve the 52m length of the tank and 192 points were used for the 0.4m height, providing a 0.006m horizontal resolution and 0.002m vertical resolution. To easily compare these numerical results to the experimental values in Grimshaw et al., our Coriolis parameter was based on their lowest presented value of $f$, which had a value of $0.105\text{s}^{-1}$. It was also decided to base the initial perturbation width $w_0$ on twice the Rossby radius of deformation ($Ro_r = \frac{U}{f}$) so as to allow for a neater examination of the parameter space. We used the same change in density as Grimshaw et al., 1% between the upper and lower fluids. In each of the simulations, the initial conditions were given by a quiescent fluid, and a

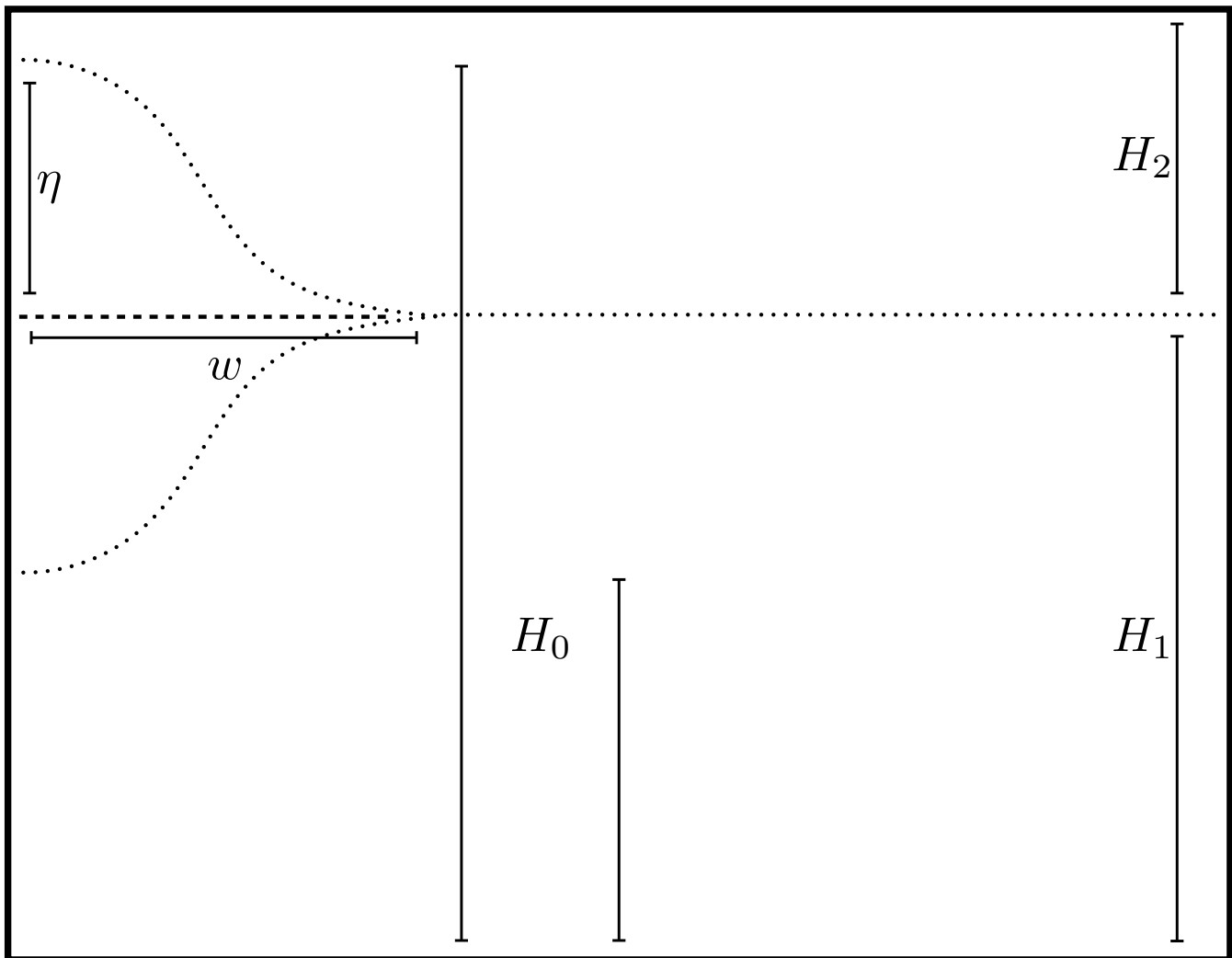

**Figure 1.** A schematic of the tank simulation set-up which illustrates the different parameters. The dotted line represents the isopycnal found at the centre of the pycnocline on the far right of the domain. The largest deflection (both polarities are shown in the figure) occurs at the left end point of the domain. $H_1$ and $H_2$ represent the depth of fluid below and above the centre of the undisturbed pycnocline, respectively. $H_0$ is the maximum or minimum height of the pycnocline created by the initial conditions. $\eta$ is the isopycnal displacement. $w$ is the width of the initial condition defined from the left-hand wall to where the pycnocline reaches within 1% of the undisturbed height.

density field defined via the isopycnal displacement $\eta$,

$$\rho'(x,z,t=0) = -0.005 \tanh\left(\frac{z - \eta - 0.3}{0.01}\right), \tag{4}$$

$$\eta = \pm 0.05 \exp\left[-\left(\frac{x}{w}\right)^8\right], \tag{5}$$

where $w$ is the half width of the initial perturbation and the sign changes correspond to changes in perturbation polarity.

## 3  Results

In this section we present the results of multiple numerical simulations. Parameters were primarily modified by changing either the initial width of the perturbation or by changing the rotation rate. Using the initial width as the typical length scale, $L = w$, we are thus varying the Rossby number. The resulting values of $Ro$ are shown in Table 1. Across all the cases, the depth does not change, and hence neither does the two-layer linear long wave speed $U = \sqrt{g\frac{\Delta\rho}{\rho_0}\frac{H_1 H_2}{H_1 + H_2}} = 0.0858\,\mathrm{m\,s^{-1}}$, where $g = 9.81\,\mathrm{m\,s^{-2}}$, $\Delta\rho = 10\,\mathrm{kg\,m^{-3}}$, $\rho_0 = 1000\,\mathrm{kg\,m^{-3}}$, $H_1 = 0.3\mathrm{m}$ and $H_2 = 0.1\mathrm{m}$. Using the same wave speed and length scales as for the Rossby number, the corresponding Reynolds numbers can be calculated, however since we are primarily concerned with internal waves, viscosity is negligible until the waves disperse to scales where viscosity is dominant, Vallis (2006). The kinematic viscosity was the same for all simulations, $\nu = 1\cdot 10^{-6}\,\mathrm{m^2\,s^{-1}}$. Several additional experiments were carried out changing the initial wave amplitude which results in different 'nonlinearity' parameters. For the initial amplitude of $\eta = 0.05\mathrm{m}$ and undisturbed pycnocline height $H_1 = 0.3\mathrm{m}$ we have $\alpha = 0.1667$. For the cases where amplitude is halved and quartered corresponding alpha values are $\alpha = 0.0833$ and $\alpha = 0.0416$. The initial value of $\alpha = 0.1667$ allows for an easy comparison to many of the figures in Kuo and Polvani which are based on a value of 0.1. In addition to the simulations above, another set of simulations were performed using the opposite polarity of the initial disturbance. These opposite polarity simulations correspond exactly to the cases seen in Table 1, the only difference being the sign in the isopycnal displacement used in the initial conditions.

Several simulations were also performed on an extra-long tank to investigate the long-time results of adjustment. For these simulations the tank length was $L = 260\mathrm{m}$ and the number of horizontal grid points was increased to 16384, providing a 0.0158m resolution. The vertical height and grid points were kept the same from the smaller case.

Unless otherwise stated the following scaling is used for all figures: $T = 1/f$, $L_z = Lz$ and $L_x = Ro_r$, with $Lz = 0.4\mathrm{m}$ corresponding to the depth of the tank. For kinetic energy, we scale by the maximum kinetic energy for all space and times *to show relative changes*.

### 3.1  The Non-rotating case

We begin by reproducing the results of the adjustment problem without rotation. The solution to this problem is well known, though we are not aware of any references that present the result in detail. We thus state the result, with a numerical example, and briefly outline the weakly nonlinear theory behind it. Non-rotating adjustment yields either a rank ordered train of solitary waves, or an undular bore forming from the initial disturbance, depending upon the polarity of the initial disturbance. Examples of these two cases are shown in Fig. 2. Since there is no rotation, the advective time-scale was chosen, $T = L/U$,

to nondimensionalize time, with the initial width $w = \frac{1}{2}w_0$ chosen for the typical length-scale. The stark difference between these cases is readily apparent in both types of plots.

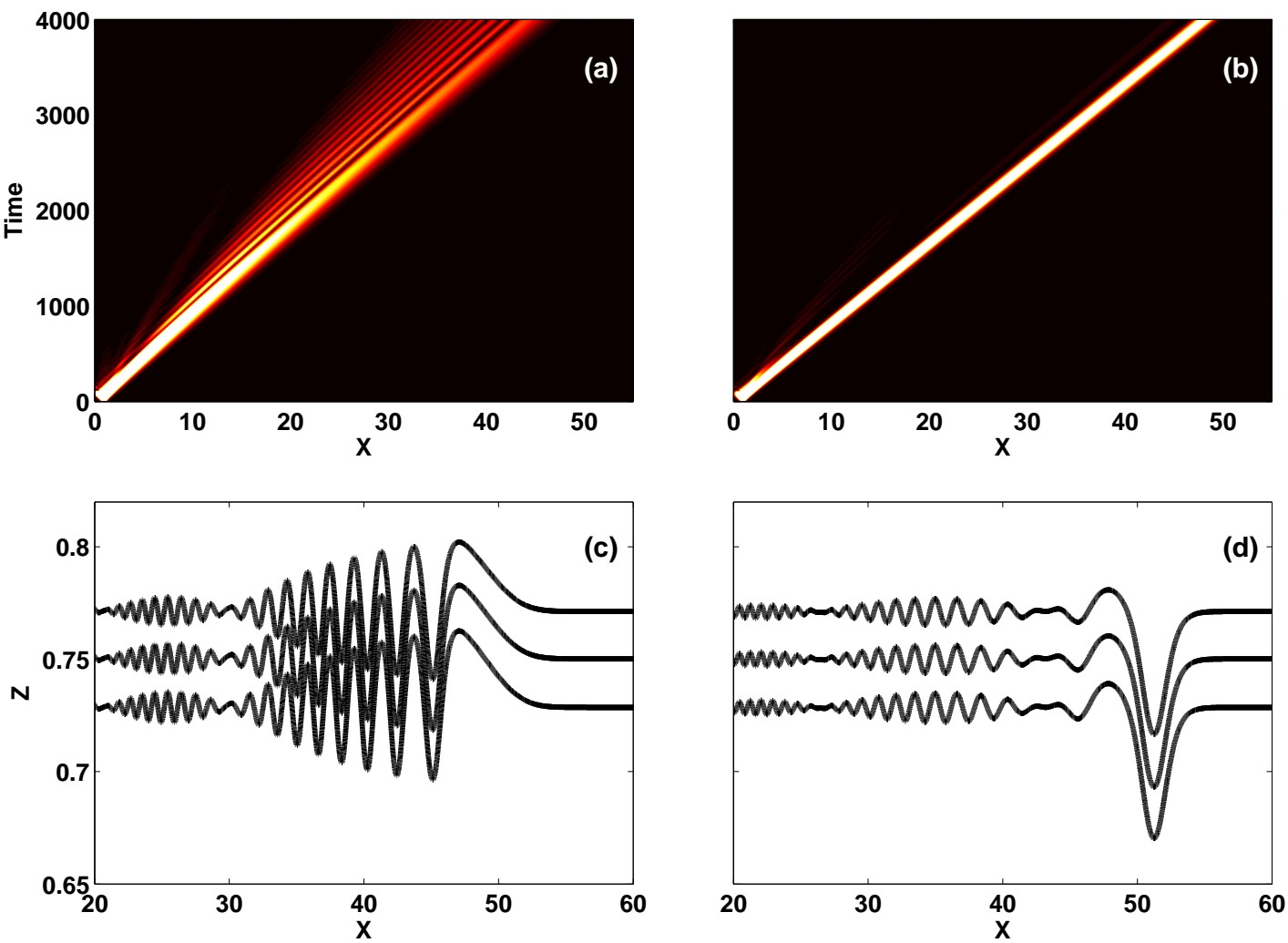

**Figure 2.** A space-time filled contour plot of vertically integrated kinetic energy and density isocontours at $t = 4275$ showing the differences between the positive and negative polarity initial conditions with $w = \frac{1}{2}w_0$. (a) and (c) correspond to the positive polarity case while (b) and (d) to the negative case.

The result may be understood in terms of KdV theory. Using the notation of Lamb (1997) separation of variables is applied to the streamfunction so that

$$\psi(x,z,t) = B(x,t)\phi(z). \tag{6}$$

The vertical structure is determined from a linear eigenvalue problem, while to first order in amplitude and aspect ratio $B(x,t)$ is governed by a KdV equation for waves propagating in each direction. The KdV equation corresponding to rightward propagating waves reads

$$B_t = -cB_x + 2cr_{10}BB_x + r_{01}B_{xxx}, \tag{7}$$

where $c$ is the linear longwave speed, $r_{10}$ is the nonlinearity coefficient, and $r_{01}$ is the dispersive coefficient. The numerical value for $c$ is computed from the linear longwave eigenvalue problem (Lamb's equation 8a), while $r_{10}$ and $r_{01}$ are computed from the integral expressions involving the eigenfunctions of the same problem (Lamb's equations 10a and 10b). The dispersive coefficient, $r_{01}$, is always negative, while the nonlinear coefficient, $r_{10}$, switches sign depending on the functional form of the stratification. In the case of a two-layer flow exact expressions can be derived. Solitary wave solutions of (7) are of the classical sech$^2$ form. The propagation speed equals the linear long wave speed to leading order, with a nonlinear correction that is proportional to amplitude and $r_{10}$ (Lamb's equation 17). Thus the sign of $r_{10}$ also determines solitary wave polarity. In the absence of background shear currents this implies that stratifications centered above (below) the mid-depth yield solitary waves of depression (elevation). All numerical experiments performed with exact solitary waves computed using the DJL equation that we are aware of match the predictions of the KdV theory presented above, as far as solitary wave polarity is concerned. Of course, KdV theory is not necessarily a quantitatively accurate predictor of the structure of large solitary waves (Lamb (1997) is one of many papers to discuss some of the discrepancies).

## 3.2 Rotation modified evolution

As discussed in the introduction, a variety of model equations have been derived that account for the effects of rotation, with Grimshaw and Helfrich (2008) providing a relatively recent summary. The essential aspects of the role of rotation can be gleaned from linear theory. In this case the streamfunction is governed by

$$\left(\mu^2\psi_{xx} + \psi_{zz}\right)_{tt} + \frac{1}{Ro^2}\psi_{zz} + \frac{1}{Fr^2}N(z)^2\psi_{xx} = 0, \tag{8}$$

where $\mu = H/L$ is the aspect ratio. When the assumption of a linear stratification is made the vertical structure of $\psi$ is sinusoidal, so that for the first vertical mode a separation of variables like (6) yields

$$\left(\mu^2 B_{xx} - \frac{\pi^2}{H^2}B\right)_{tt} - \frac{\pi^2}{H^2}\frac{1}{Ro^2}B + \frac{1}{Fr^2}B_{xx} = 0. \tag{9}$$

The well know dispersion relation of rotation modified internal waves in a channel is readily recovered by assuming a traveling wave solution Vallis (2006). In the hydrostatic limit this equation reduces to the simplest example of a partial differential

equation that is both hyperbolic and dispersive; the classical Klein-Gordon equation of mathematical physics,

$$B_{tt} + \frac{1}{Ro^2} B = \frac{H^2}{\pi^2 Ro^2} B_{xx}. \tag{10}$$

By using the plane wave anstaz, it can immediately seen that the dispersion relation yields a lower bound on frequency in the long wave limit, and hence the phase speed is unbounded in this same limit. This is the central problem that model equation theories such as the Ostrovsky equation face when implemented numerically. It is also readily apparent from (10) that a non-trivial, time independent state is possible, and this state corresponds to the geostrophic state of classical geophysical fluid dynamics Vallis (2006). Furthermore, it is clear that a spatially independent inertial oscillation is a possible solution to the equation. However, for a given initial condition it is not immediately obvious what the precise split is between the portion of the initial state that propagates away and the portion left behind. While the case in which the disturbance that emanates from the initial condition is small enough to be well described by linear wave theory has been studied in detail by Lelong and Sundermeyer Lelong and Sundermeyer (2005), the significant amount of literature on the combined effects of nonlinearity, dispersion and rotation, and especially the experimental results in Grimshaw et al. (2013), suggest that the initial value problem should be reconsidered without *a priori* approximations.

Using our definition of the Rossby number, we find that for the experiments presented in Grimshaw et al. (2013), with a Coriolis parameter of $f = 0.105\text{s}^{-1}$, had a corresponding Rossby number of $0.667$. Therefore, we consider the $f = f_0$ and $w = w_0$ case ($Ro = 0.5$) as our baseline. We also pick a negative initial condition to match their configuration. With this in mind, Fig. 3 compares this baseline case with one where the only difference is that the rotation rate has been quartered ($f = \frac{1}{4}f_0$, $Ro = 2$). Panels Fig. 3 (a) and Fig. 3 (c) correspond to the baseline case, while panels Fig. 3 (b) and Fig. 3 (d) correspond to the reduced rotation case. Panels Fig. 3 (a) and Fig. 3 (b) show vertically integrated kinetic energy space-time plots, while panels Fig. 3 (c) and Fig. 3 (d) show density isolines at $t = 47.25$ and $t = 11.81$ respectively. The vertical lines correspond to the locations of the waves that emanate from the initial disturbance as described by linear theory. We computed the spectrum of the horizontal velocities to extract the dominant wavenumbers ($k \approx 0.84$ and $k \approx 0.48$ respectively) and used the algorithm outlined in Stastna and Rowe (2007) to calculate the speeds. We have presented both the linear phase and linear group speeds.

Comparing the results seen in Fig. 3 with Fig. 2 (b) and (c) (since they both began with a negative polarity initial condition), there are immediate differences in both styles of plots. The most striking of these differences are the retention of energy in the geostrophic state, and the spreading of the ejected waves. The geostrophic state is visible in all plots along the left-hand of the tank (near the wall). Comparing the two columns in Fig. 3, the case with a stronger rotation rate traps more energy in the geostrophic state. We will investigate differences in the geostrophic state in section 3.4. The wave spreading is visible in the space-time plots as the waves propagate and within the new structure of the density isolines. At both rotation rates, the solitary wave, which is produced in the non-rotating case, has broken down into a series of smaller waves. A transition also appears to occur in the wave speed as the rotation rate changes. In Fig. 3 (d) the wave front roughly corresponds to the linear phase speed (which is, in turn, a good approximation to the solitary wave propagation speed), while in Fig. 3 (c) the front appears to have shifted to the linear group speed. This suggests that the low rotation case develops a wave train that can be interpreted

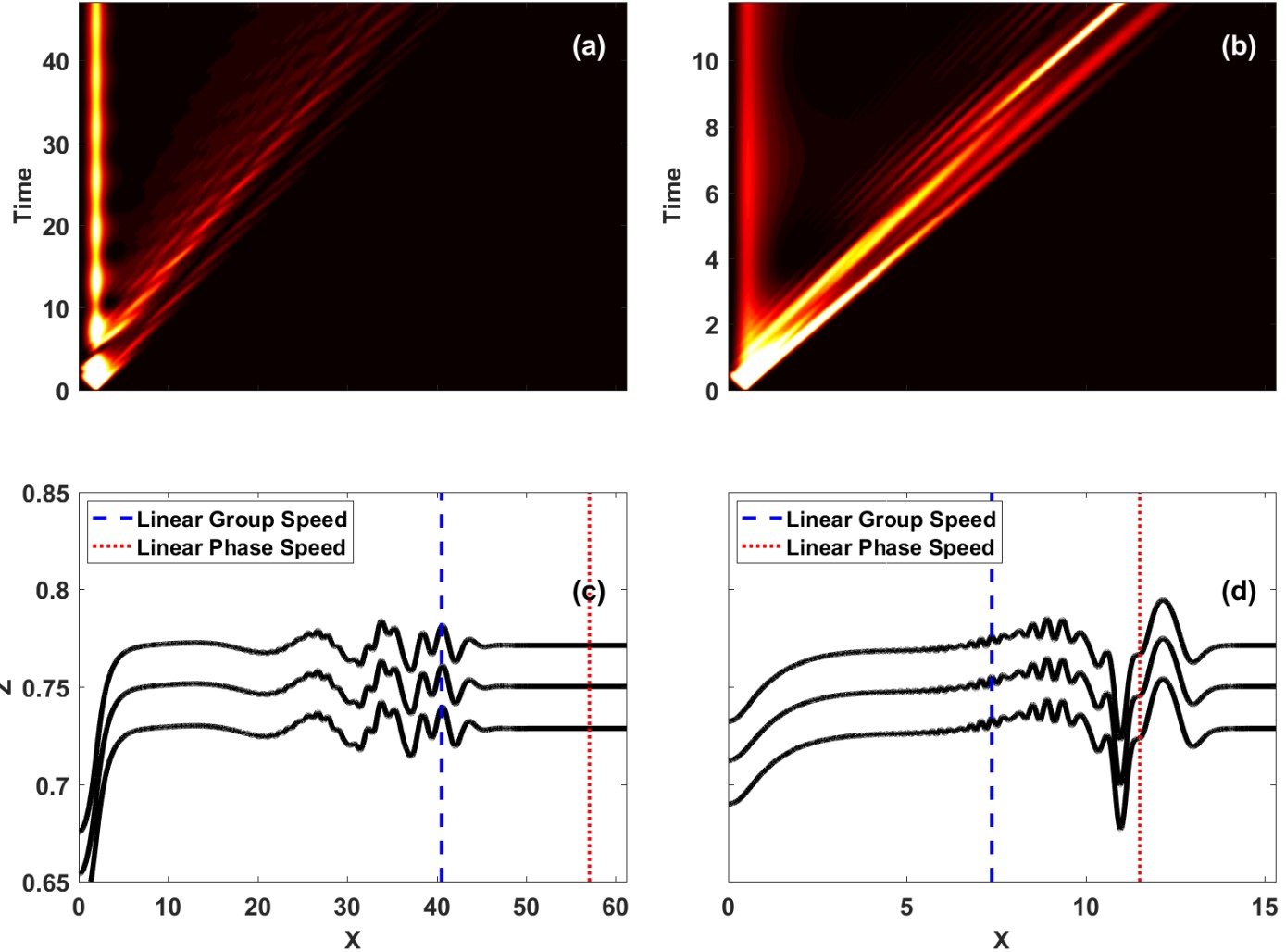

**Figure 3.** Fig. 3(a) and Fig. 3(b) show a space-time plot of vertically integrated kinetic energy while Fig. 3(c) and Fig. 3(d) show three density isolines at $t = 47.25$ and $t = 11.81$ respectively. Figure 3(a) and Fig. 3(c) correspond to the $Ro = 0.5$, $f = f_0$ and $w = w_0$ case, and Fig. 3(b) and Fig. 3(d) to the $Ro = 2$, $f = \frac{1}{4}f_0$ and $w = w_0$ case. The vertical lines in Fig. 3(c) and Fig. 3(d) represent the distance the waves would have travelled according to the linear phase and group speed.

as rotation modified solitary wave (at least on the time scales considered) while the high rotation case develops a wave train that can be interpreted as a wave packet. The importance of nonlinearity for both of these cases, and indeed for the geostrophic state remains to be assessed.

Observing the structure that appears throughout the figure, we argue that these waves closely resemble a modulated wave packet as presented in Grimshaw et al. (1998), instead of the rotation modified bore seen in Kuo and Polvani (1997). When comparing to the work done by Kuo and Polvani, we first note that for our simulations the nonlinear parameter is quite small at $\alpha \approx 0.166$. However, their work suggests that even for this small value and smooth initial condition breaking will still occur.

The present simulations were carried out with the full set of incompressible Navier-Stokes equations. As such, the dispersion that is neglected in the shallow water equations, used by Kuo and Polvani, becomes important when the wave front steepens. Dispersion breaks the front down into a train of smaller waves and eliminates shock formation. In addition to the change in steepening dynamics, the initially localized waves disperse over time, yet are observed to remain bound together (corresponding to the width of the packet envelope). For this reason we find that the modulated wave packet is a better description for these

dynamics, though we note that in all our simulations the wave packet never completely separates from the trailing waves. This description is also supported by the shift in propagation speed to the linear group speed, as this is the first order estimate of the speed which a wave packet would propagate at.

### 3.3 Nonlinear and polarity effects

Since the majority of classical literature on the geostrophic adjustment problem considers the linear problem, it is important

to clearly identify those aspects of our simulations that are nonlinear in nature. One way to investigate the nonlinear effects in the evolution, shown in Fig. 3, is to consider how the spectrum evolves in time, since (in the absence of dissipation) linear dispersive waves maintain the spectrum of the initial conditions for all times. Fig. 4 shows the spectrum of the horizontal velocity profile at the surface (the results at other depths, and indeed for other fields yielded qualitatively unchanged results) at various times for both cases shown in Fig. 3. Respectively, these correspond to $t = 15.75$ and $t = 3.94$, $t = 31.50$ and $t = 7.87$,

$t = 47.25$ and $t = 11.81$, and $t = 63$ and $t = 15.75$. The spectral power density was scaled by the maximum power for all profiles shown in order to highlight the differences. It is readily apparent from Fig. 4 (a) that in the $Ro = 0.5$, $f = f_0$ and $w = w_0$ case there is little change in the spectrum as time evolves. As time increases there appears to be a slow decay in the power at the excited wavenumbers. There is no shift in wavenumber for the various peaks, or indeed any other major change in the spectrum evident. This is not the case in Fig. 4 (b) for the $Ro = 2$, $f = \frac{1}{4}f_0$ and $w = w_0$ case. The spectrum in this case,

contains large fluctuations (more than $25\%$ for the peak value) in power, and shifts in the excited wavenumbers. These changes in the spectra as time evolves are hallmark effects of nonlinearity, and hence indicate that the while the emanating wavetrain in the $Ro = 0.5$ case appears to be well described by linear theory, if we consider weaker rotation effects such as in the $Ro = 2$ case, linear theory is no longer a useful description.

In order to investigate these effects in a more systematic manner, we started from the case with $w = \frac{1}{2}w_0$ (which was the

smallest width that still produced a solitary wave) and ran several simulations where we varied the amplitude (by halving it), varied the rotation rate (which was quartered to show clear differences), compared the different polarities and considered an extremely small amplitude 'nearly linear' case. To compare with known nonlinear wave results we computed the non-rotating version of a number of the cases. A comparison of the 1D-averaged KE for a number of the cases is presented in Fig. 5, where we have scaled any reduced amplitude cases so that, were linear theory to apply, the curves would collapse onto a

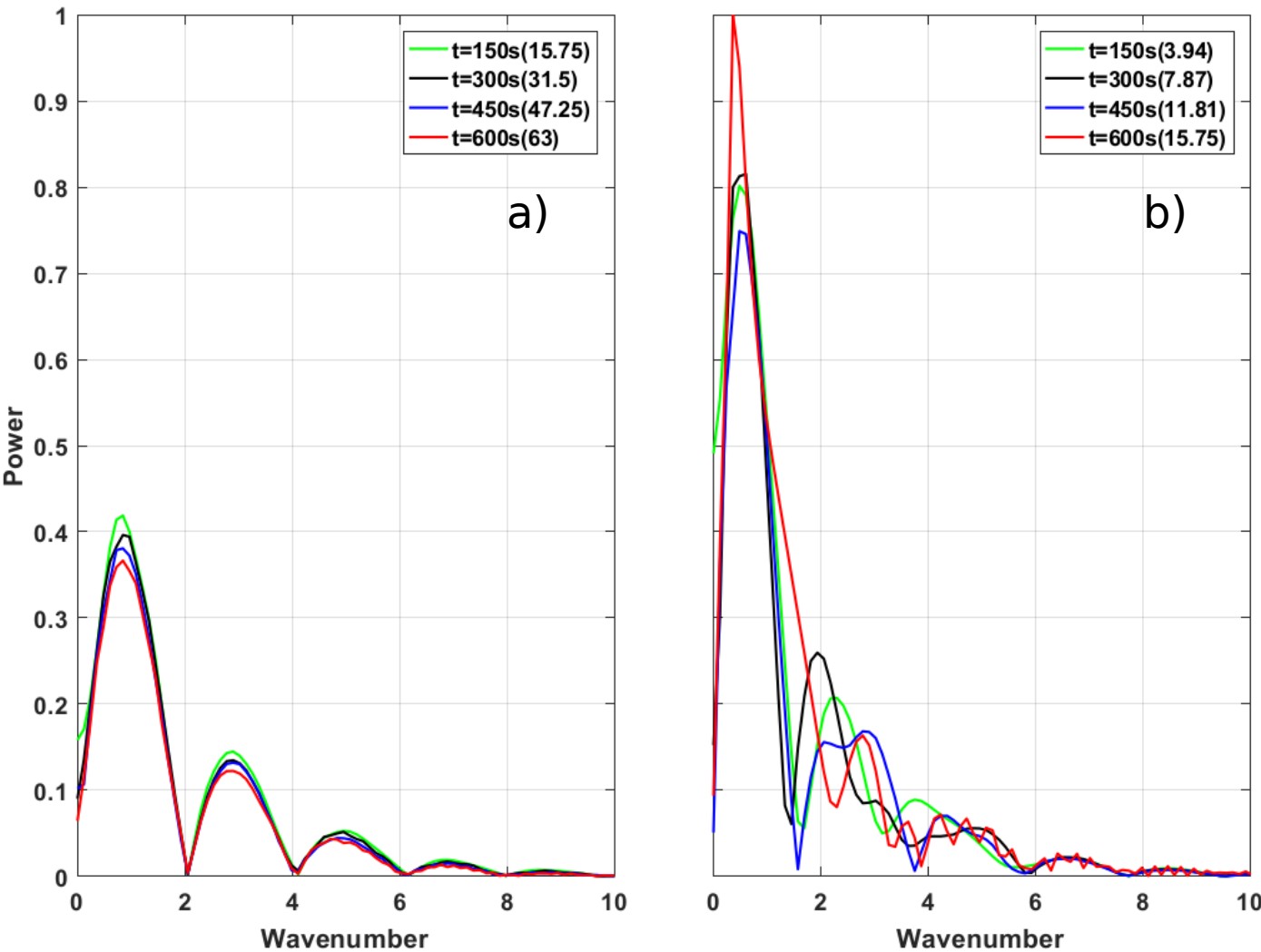

**Figure 4.** A comparison of how the horizontal velocity spectrum for the $Ro = 0.5$, $f = f_0$ and $w = w_0$ case (a), and the $Ro = 2$, $f = \frac{1}{4}f_0$ and $w = w_0$ case (b) change over time. The spectral power has been scaled by the maximum between the cases to highlight the differences. The green line corresponds to $t = 15.75$ and $t = 3.94$, the green line corresponds to $t = 31.50$ and $t = 7.87$, the blue line to $t = 47.25$ and $t = 11.81$, and the red line to $t = 63$ and $t = 15.75$. There are clear changes in the spectra for the weakly rotating case as the waves evolve. In the strong rotation case the only temporal differences are a slow decay. The differences seen in the weakly rotating case highlight the nonlinear effects which are present in this regime.

single profile. Kinetic energy was chosen as the variable shown, since it provides information about both the structure of the dynamics and magnitude of the velocities. Fig. 5 (a) shows how the non-rotating adjustment yields waves that are profoundly affected by changes in polarity, and to a lesser degree by changes in amplitude. A solitary wave train is observed for negative polarity case, and an undular bore for the positive polarity case. The change in amplitude results in a phase shift, however the amplitude of the solitary wave remains nearly constant. These results are a clear indication of nonlinear behaviour for the non-rotating case. In Fig. 5 (b) we compare several cases with rotation following a similar methodology as Fig. 5 (a), however we have included our 'nearly linear' case where the amplitude has been reduced by a factor of 200. The change in polarity does not significantly change the dynamics of the ejected waves, with the largest change between these cases being that the positive polarity case has a higher amplitude both within the wave packet and in the geostrophic state. The effect of changing the amplitude does not significantly change the wave packet, since the wave packet is quite small in this case, and hence to leading order can be understood from the point of view of linear dispersive wave theory, similarly to what was observed based on the spectrum in the discussion above (Figure 3 and the related discussion). For the geostrophic state, the changes in the initial disturbance amplitude result in changes to the amplitude and the location of the peak in kinetic energy, with the reduction in amplitude yielding a greater than linear response in the amplitude of the geostrophic state. Again, changing polarity yields the most significant changes. Linear theory, as exemplified by the green curve, provides a reasonable prediction, though details are amplitude dependent. For Fig. 5 (c) we kept the polarity of the initial disturbance negative and compared the change in amplitude for a smaller rotation rate, as well as the non-rotating case. The lower rotation rate allows for more energy to be deposited into the wave train. The primary change for the reduction in amplitude is that the individual waves within the wave packet of the scaled reduced case (in blue) appear to be larger in amplitude compared to the base case (in black). There also appears to be a slight phase shift between the cases (consistent with a packet that travels at a slightly different speed). Comparing these two cases to the non-rotating case shows that while the peak in energy has been shifted back, the wave front of the solitary wave and the wave packets are at roughly the same location. For this lower rotation case there is very little difference in the geostrophic state as a result of amplitude reduction, implying that for low rotation rates the geostrophic state can be well-described by linear theory.

To investigate the nonlinear effects that arise from changes in polarity in the geostrophic state (Fig. 5 (b)), long-time simulations with a rotation rate set to $f = 2f_0$ and an initial width of $w = \frac{1}{2}w_0$ were computed, resulting in a Rossby number of $\frac{1}{2}$. These results are presented in Fig. 6. Fig. 6 (a) and Fig. 6 (b) show the vertically integrated kinetic energy at the location of the maximum induced by the geostrophic state. Fig. 6 (c) and Fig. 6 (d) show the total spatial distribution of the normalized kinetic energy in the region around the geostrophic state at 720 s (151.2), along with three contours of constant density, for a negative and positive initial polarity respectively. Fig. 6 (a) clearly shows the energy difference that was seen in Fig. 6 (b), indicating that the positive polarity case appears more efficient at keeping energy in the geostrophic state. Fig. 6 (b) shows the time series of the logarithm of kinetic energy after the packet has been ejected. If we ignore the inertial oscillations, which appear to be rapid on the time scale shown, we can see a clear decay. From this panel it is also possible to note that the oscillations appear to persist significantly longer in the positive case. By computing the logarithm of the time series (Fig. 6 (b)) we are able to show that the decay is nearly exponential, with the positive polarity case decaying roughly 5% faster. The decay rate decreases

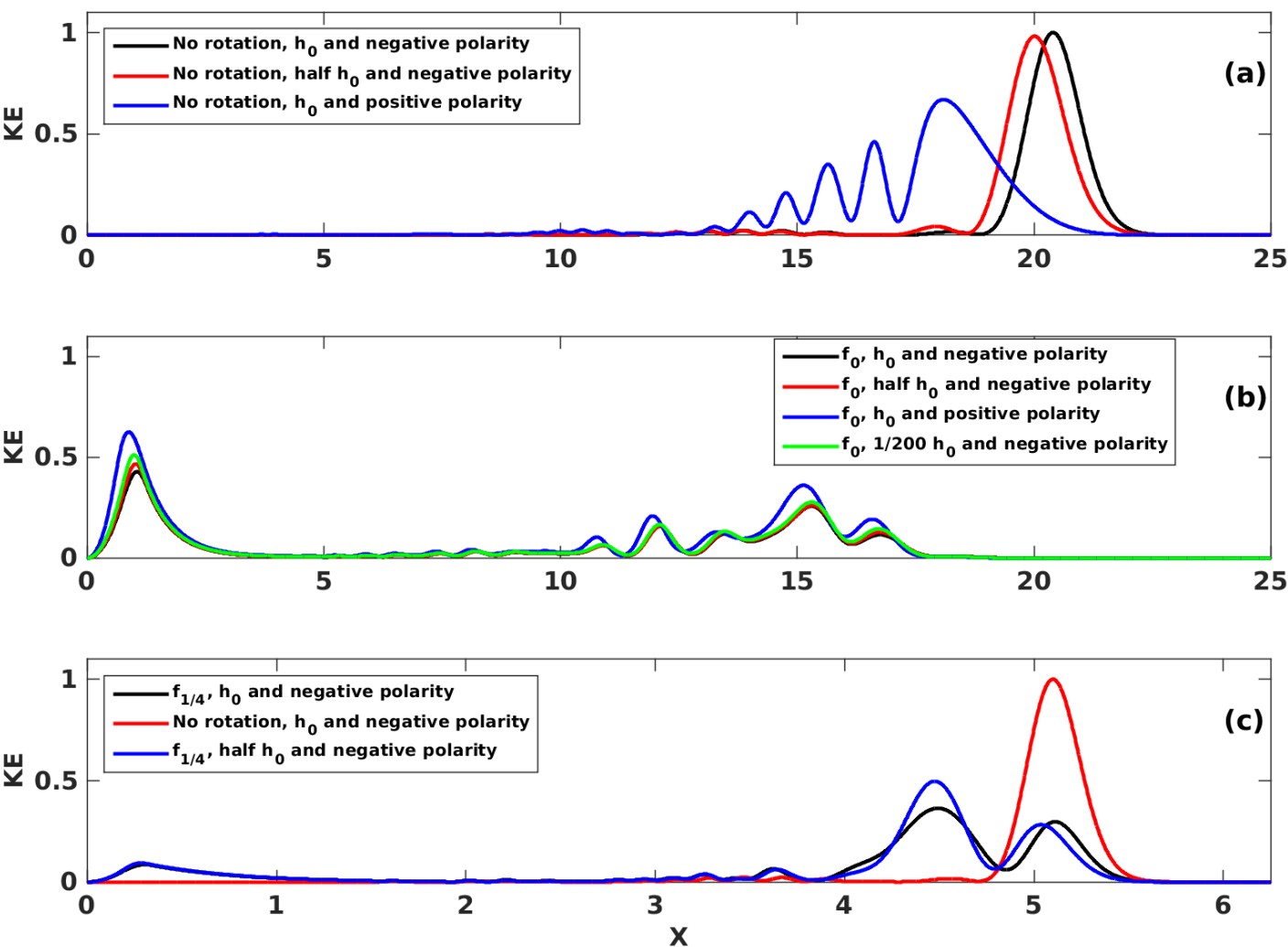

**Figure 5.** A comparison of the 1D KE for several different cases to outline the effects of non-linearity. All plots are taken at $t = 1140$ for the non-rotating case, $t = 25.2$ for the $f_0$ case ($w = \frac{1}{2}w_0$, $Ro = 1$), and $t = 6.3$ for the $\frac{1}{4}f_0$ case ($w = \frac{1}{2}w_0$, $Ro = 4$). The reduced amplitude nearly linear cases have been scaled by the change in amplitude squared. The kinetic energy has also been normalized by the maximum of the non-rotating negative polarity case. (a) highlights the differences that arise from halving the initial amplitude and by changing the polarity of the non-rotating case. (b) compares the same changes as the panel above, however we have included rotation, and also consider a 'nearly linear' case with an initial amplitude of $1/200\eta_0$. (c) is a comparison of different negative polarity cases, one with no rotation and the standard amplitude, and two others at $\frac{1}{4}f_0$ with standard and half amplitude.

over time. From the bottom two panels, Fig. 6 (c) and Fig. 6 (d), it is clear that the polarity of the geostrophic state strongly modifies the vertical distribution of the kinetic energy. Note in particular the difference in strength of kinetic energy below the pycnocline, and the tilt of the high kinetic region that follows the deformed pycnocline.

To quantify the nonlinear behaviour of the geostrophic state and the inertial oscillations that accompany it, Figure 7 shows the differences in total kinetic energy within the geosotrophic state between our original amplitude cases and cases with a ten-fold reduction reduction in amplitude. As in Fig. 5 we have scaled up the reduced amplitude cases and thus for a purely linear problem there should be no differences between the two curves shown. Figure 7 shows five cases of differing initial width and the same Coriolis parameter $f = f_0$, Fig. 7 (a) to $w = \frac{1}{4}w_0$ ($Ro = 2$), (b) to $w = \frac{1}{2}w_0$ ($Ro = 1$), (c) to $w = w_0$ ($Ro = 0.5$), (d) to $w = 2w_0$ ($Ro = 0.25$), and (e) $w = 4w_0$ ($Ro = 0.125$). Though the nearly linear case does behave in a qualitatively similar manner to the original cases there are key differences. For one, in Fig. 7 (a)-(c) there are clear magnitude differences between the two cases. For almost all times the scaled kinetic energy curve for the amplitude reduced case lies below the corresponding curve for the original cases. This remains true for the later cases ((d) and (e)), though not to the same extent. Second, closely examining the inertial oscillations reveals that they appear to decay more rapidly for the amplitude reduced cases compared to the original cases, this can especially be seen in Fig. 7 (c) and (d). Only for the two rightmost panels ($Ro \leq 0.25$) could it be said that the two curves are nearly coincident. Thus even the geostrophic state exhibits clear evidence of nonlinearity.

The primary dynamic variable for these simulations is the Rossby number since both changes to rotation rate and changes to the initial width are both just modifications to this dimensionless parameter. A different manner in which the effects of nonlinearity may be investigated, is by asking if the dynamics collapse onto a single case for the same Rossby number, Fig. 8 (a)-(c) show the space-time plots of vertically integrated kinetic energy for three cases with the same Rossby number but different combinations of parameters. The Froude number, $Fr = U/c$, is computed dynamically, with $U$ set by the maximum horizontal velocity from the simulation at a given time, and $c$ given by the linear group speed calculated using the algorithm outlined in Stastna and Rowe (2007) using the dominant wavenumbers ($k \approx 0.84$ and $k \approx 0.48$ respectively) for each rotation rate. Fig. 8 (a) corresponds to $f = f_0$ and $w = \frac{1}{2}w_0$, Fig. 8 (b) to $f = \frac{1}{2}f_0$ and $w = w_0$ and Fig. 8 (c) to $f = \frac{1}{4}f_0$ and $w = 2w_0$. The axis of Fig. 8 (b) and (c) have been scaled by the corresponding change in Coriolis parameter ($\frac{1}{2}$ and $\frac{1}{4}$ respectively). Figure 8 (d) shows the time series of the Froude number, $Fr = U/c$. While the oscillations of the geostrophic state near $x = 1$ in the space-time plots are quantitatively similar, the number and shape of the waves that are produced in the wave train are slightly different. The reason for these differences is highlighted in Fig. 8 (d) where the Froude numbers match for early times, but begin drift rapidly. These results again illustrate the importance of nonlinear effects within this system, and the necessity to include such effects when modeling the system.

During the analysis of the numerical experiments that varied the width of the initial condition, and interesting observation about multiple wave trains was made. The initial condition yields both rightward and leftward propagating waves. For narrow initial conditions the leftward propagating waves reflect from the left wall early in the simulation and are difficult to disentangle from the initially rightward propagating wave train. However, for wider initial conditions the leftward propagating waves must travel a longer distance before reflecting off the wall, allowing for them to appear separate from rightward propagating waves. This interaction is shown in Fig. 9 using the potential energy field because the amount of spanwise velocity created in the

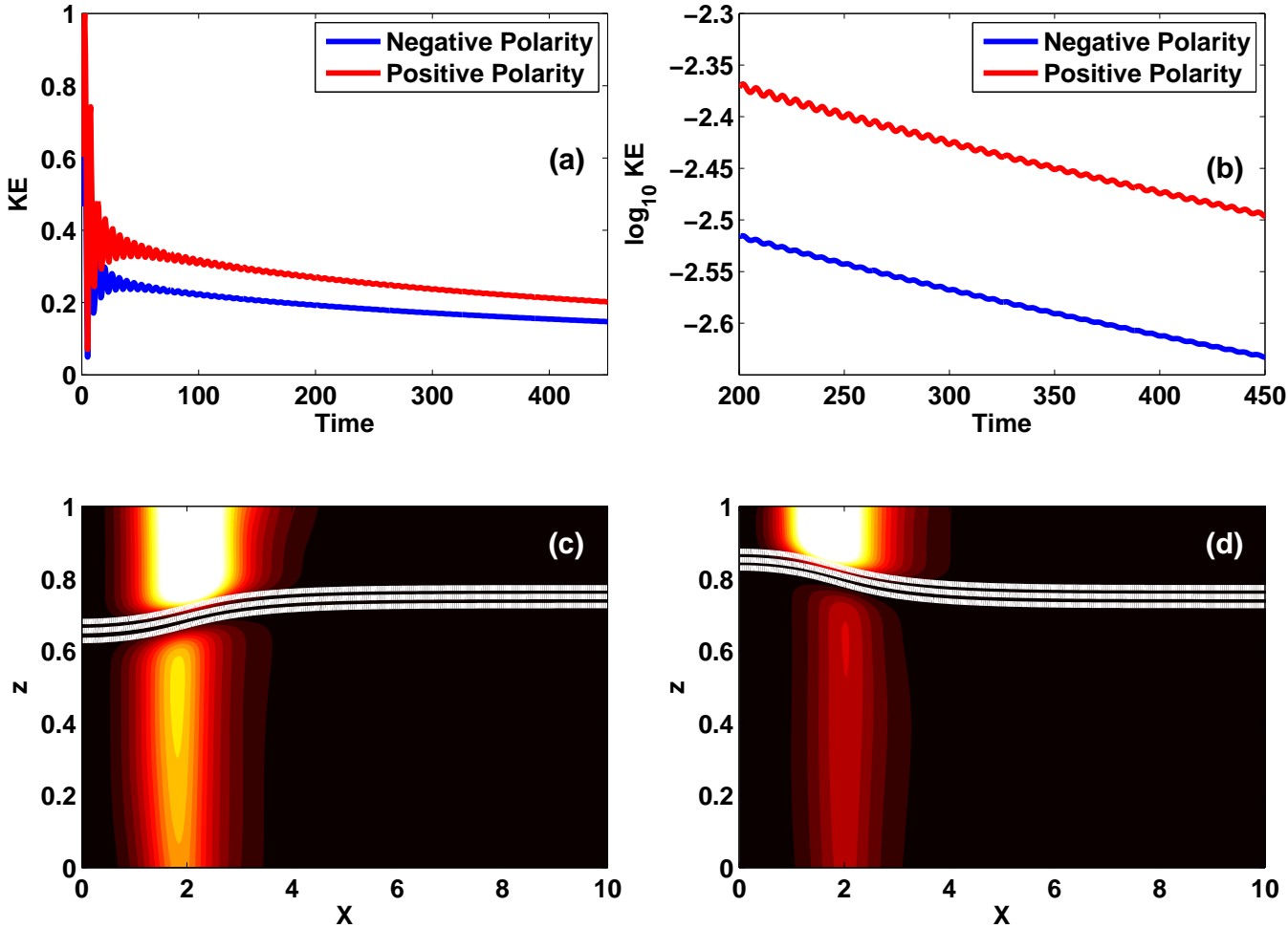

**Figure 6.** Long time simulations comparing the differences in the geostrophic state for negative and positive initializations. For both simulations $f = 2f_0$, $w = \frac{1}{2}w_0$ and $Ro = \frac{1}{2}$. (a) presents the long-time time-series of vertically integrated kinetic energy in the geostrophic state for both cases. (b) shows the base ten logarithm of the geostrophic state kinetic energy for both cases after the packet has been ejected. (c) and (d) show the shaded distribution of kinetic energy in the geostrophic state along with contours of constant density, for a negative and positive initial polarity respectively.

geostrophic state is so great that it drowns out this reflection signal in the kinetic energy field. Both cases maintain the same Coriolis parameter $f = f_0$ but Fig. 9 (a) corresponds to $w = 0.5w_0$ while Fig. 9 (b) corresponds to $w = 2w_0$. In Fig. 9 (a) it

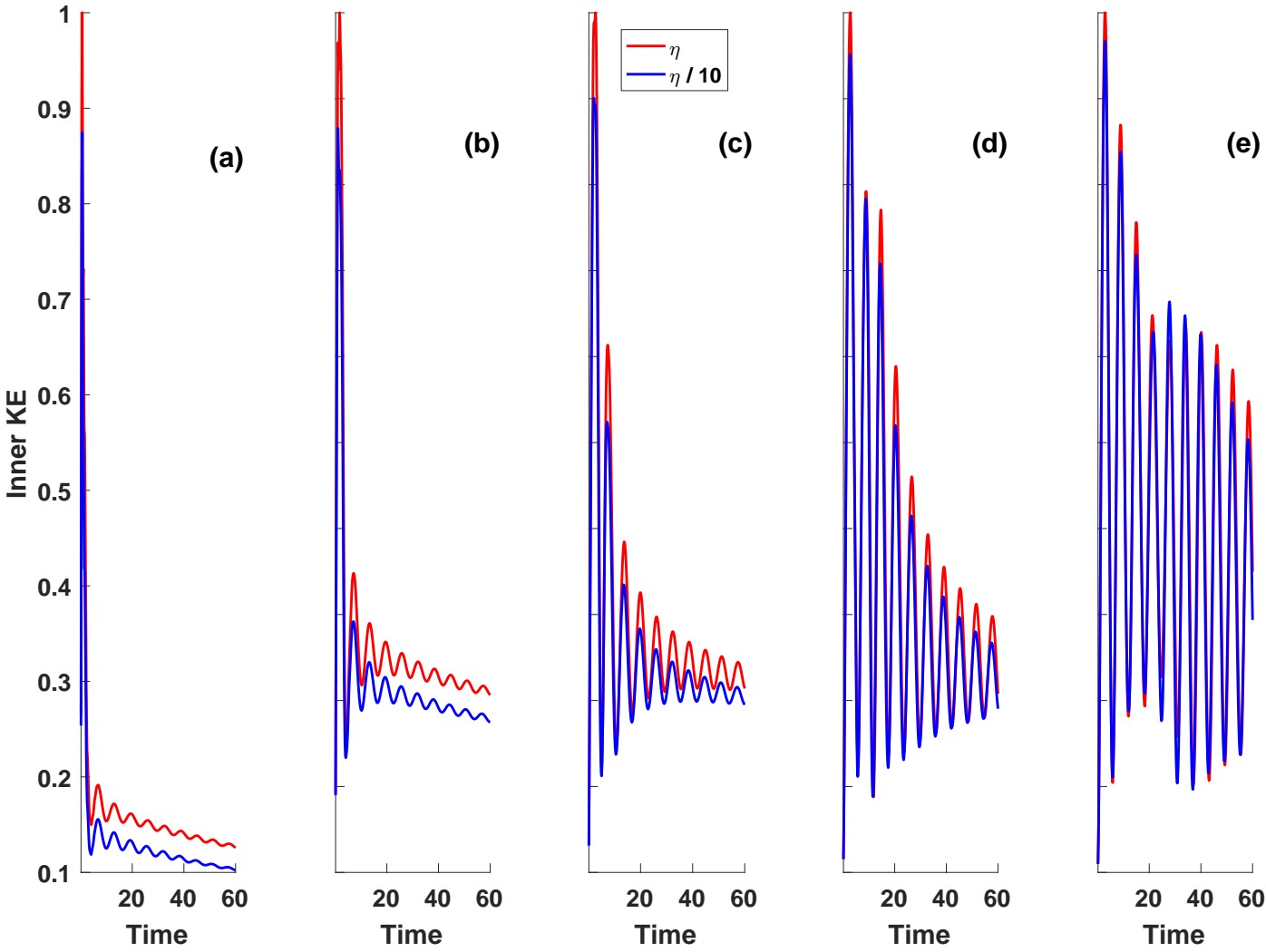

**Figure 7.** The difference in total inner kinetic energy between our original amplitude cases and cases where the amplitude has been reduce by a factor of 10. The reduced cases have then been linearly scaled to account for this amplitude change. The original energies are shown in red, while the reduced ones are in blue. Discrepancies between the two cases are due to nonlinear effects. (a) $w = \frac{1}{4}w_0$ ($Ro = 2$), (b) $w = \frac{1}{2}w_0$ ($Ro = 1$), (c) $w = w_0$ ($Ro = 0.5$), (d) $w = 2w_0$ ($Ro = 0.25$), and (e) $w = 4w_0$ ($Ro = 0.125$)

is difficult to distinguish the two wave trains (though we have superimposed colored arrows in order to accentuate the pattern for the reader). This distinction is much clearer in Fig. 9 (b) where the leftward travelling wave takes roughly twice as long to reach the left wall. In this case it is possible to distinguish the wave trains within the pattern of waves that are produced.

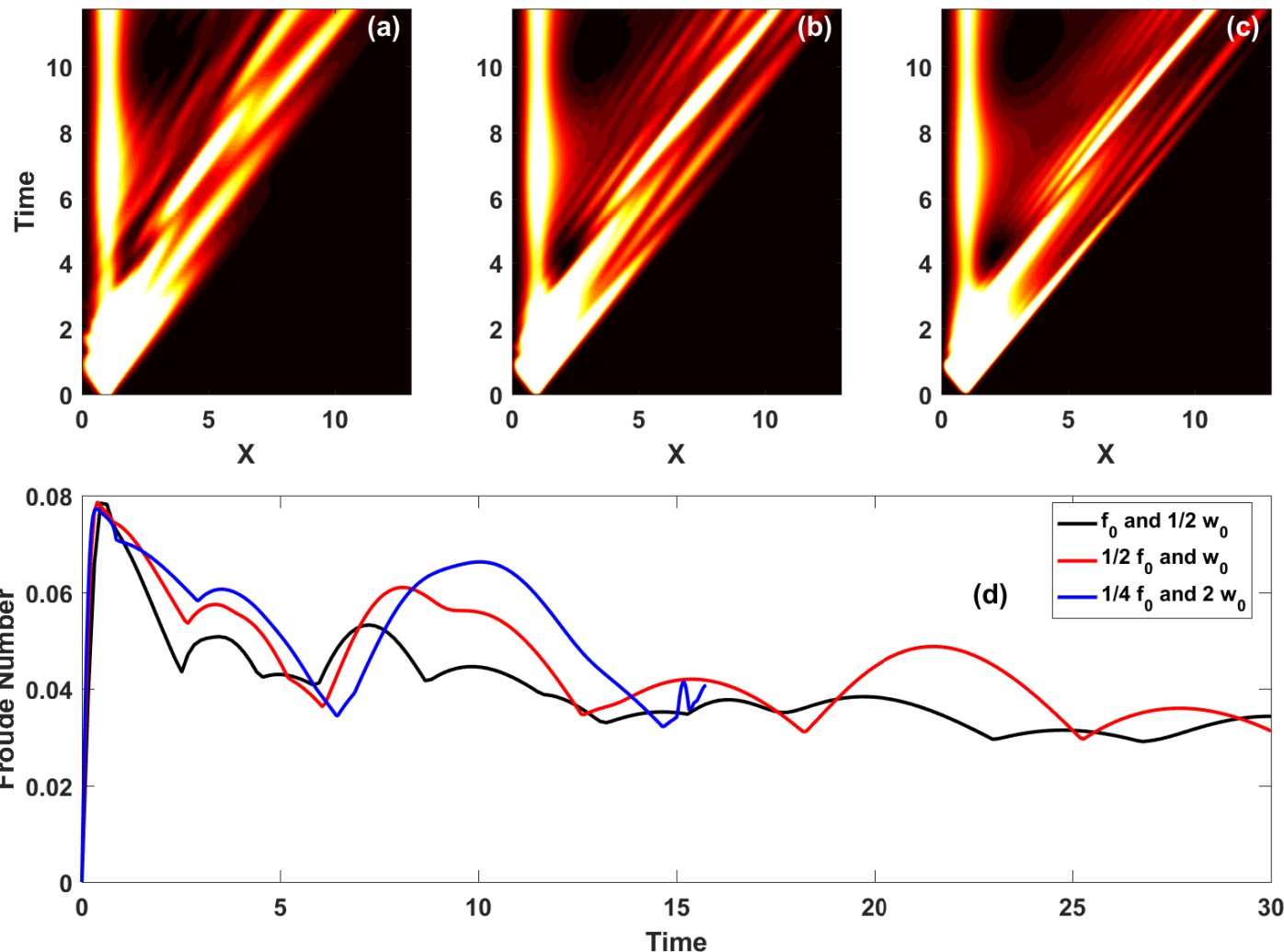

**Figure 8.** A space-time plot of kinetic energy. The different columns correspond to different combinations of $f$ and $w$ used to form a value of $Ro = 1$. (a) $f = f_0$ and $w = \frac{1}{2}w_0$, (b) $f = \frac{1}{2}f_0$ and $w = w_0$ and (c) $f = \frac{1}{4}f_0$ and $w = 2w_0$. The second row corresponds to the same case as the first columns but the aspect ratio has been scaled by the change in rotation rate.

Thus a natural method of generation of waves in a tank will create waves in both directions which must be accounted for in the interpretation of physical experiments.

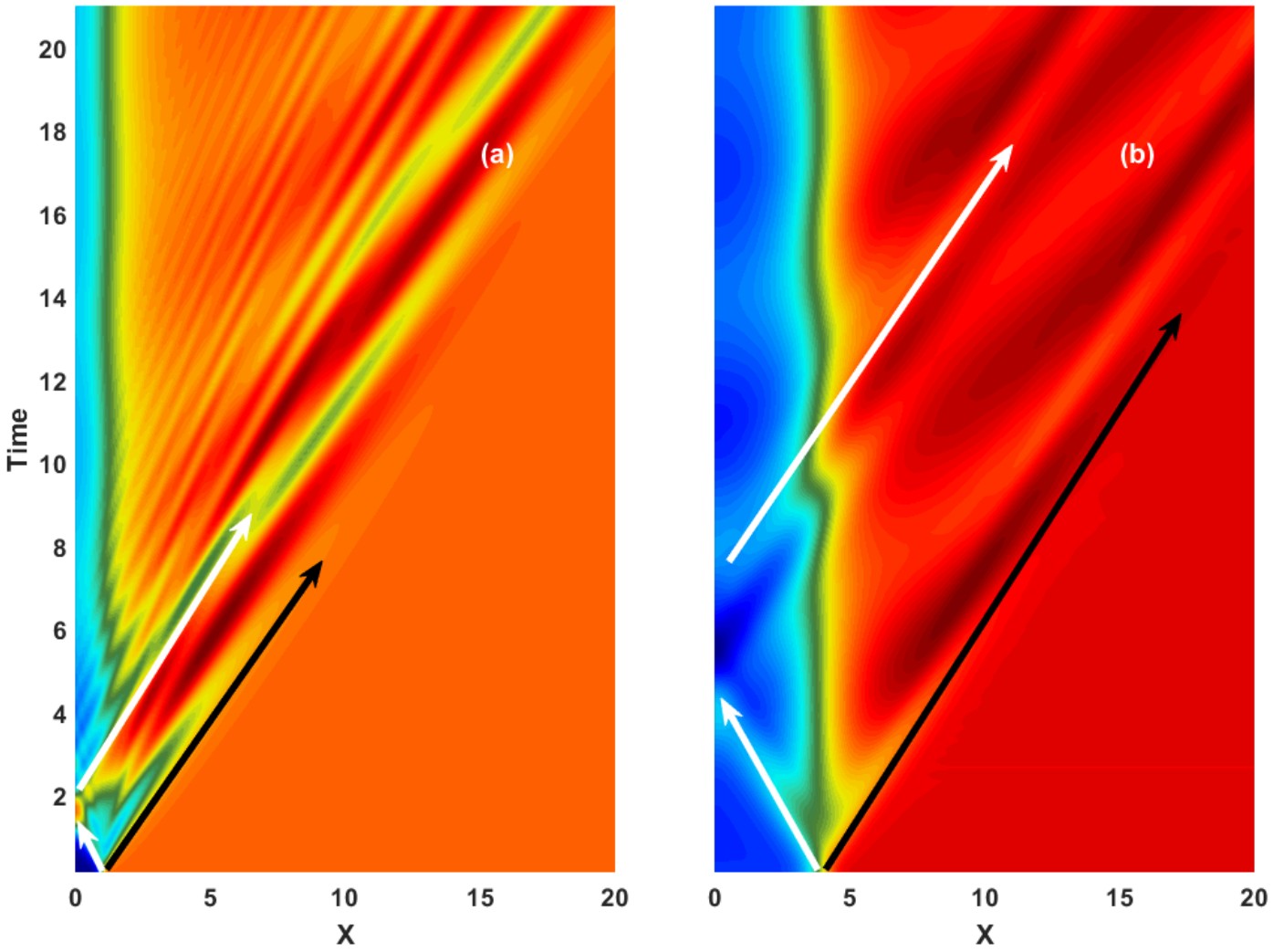

**Figure 9.** Space-time pseudocolour plots of the change in potential energy for the $Ro = 1$, $f = f_0$ and $w = 0.5w_0$ case (a), and the $Ro = 0.25$, $f = f_0$ and $w = 2w_0$ case (b). As for the kinetic energy, it has been scaled the maximum value. Visible in both cases (though significantly easier to see in (b)) there is both a rightward (black arrow) and leftward (white arrow) propagating wave train created by the initial conditions. The leftward travelling wave train will eventually reflect off the close left hand wall and propagate rightwards. If the initial conditions are quite narrow the leftward propagating wave train reflects quickly off the wall and is difficult to disentangle from the rightward propagating wave train.

## 3.4 The geostrophic state

In the rotation modified adjustment problem there are two dominant features, the geostrophic state that is left over from the initial conditions and the train of Poincaré waves which carries energy away from it. For this section we will focus on the dynamics, and changes, of the geostrophic state. We will primarily be comparing our results with those from Lelong and Sundermeyer (2005), who performed three-dimensional numerical simulations of the adjustment problem. Lelong and Sundermeyer focused on the energetics of the geostrophic state that are generated by a well mixed region of intermediate density fluid. Though not exactly the same case, Lelong and Sundermeyer performed their simulations using the full set of equations and thus provide an apt point of comparison. To facilitate this comparison between our work and theirs, Table 2 provides a translation of our notation, to that of Lelong and Sundermeyer.

A major difference between the two sets of experiments is the background stratification and density anomaly. As given explicitly in section 2, we have a two layer stratification given by the tanh function with an anomaly also given by the tanh function. Lelong and Sundermeyer use a localized anomaly diffusivity to create a two-lobed axisymmetric lens density perturbation, with a linear background stratification.

Figure 10 shows space-time plots of vertically-integrated kinetic energy within the geostrophic state for five cases of different initial widths where the rotation rate has been held constant at $f = f_0$. These cases have Rossby numbers 2, 1, $\frac{1}{2}$, $\frac{1}{4}$ and $\frac{1}{8}$ for Fig. 10 (a), Fig. 10 (b), Fig. 10 (c), Fig. 10 (d), and Fig. 10 (e), respectively. The figure has been saturated by the maximum kinetic energy across all cases. Once $Ro \geq 1$ (Fig. 10 (b)-(e)), the geostrophic state shows clear oscillations within the kinetic energy. These spikes in kinetic energy occur during the vertical oscillations of the geostrophic state. It is also possible to see from this figure that all the cases initially spike with roughly the same magnitude of kinetic energy, but then differ greatly depending on the Rossby number. In Fig. 10 (d) and (e) it is possible to identify the reflected wave interfering with the oscillations of the geostrophic state, matching the features seen in Fig. 9.

To compare the energy within the geostrophic state between cases, and with published literature, we horizontally integrate the geostrophic state (the region shown in Fig. 10) to produce a time-series of both the kinetic and potential energies. Following what was done in Kuo and Polvani, we compute the difference in these energies compared to the initial state. The results of this are shown in Fig. 11. The extent of the geostrophic state is defined as twice the distance from the left-hand wall to the maximum in kinetic energy. Due to the nature of our initial conditions, namely that we start with a smooth transition and still fluid, the ratio $\Delta KE / \Delta PE$, that is used in Kuo and Polvani, is difficult to use since the potential energy may be zero if it reaches its initial state during oscillation. While it would be possible to use the reciprocal of the ratio, we have chosen to present the differences separately ( Fig. 11 (a) for $\Delta PE$ and Fig. 11 (b) for $\Delta KE$). The energy in both plots has been scaled by the base case of $f = f_0$ and $w = w_0$. As in the results presented in figure 8 of Lelong and Sundermeyer (2005), for larger initial widths (smaller Rossby numbers) there are correspondingly larger oscillations in both potential and kinetic energy. Our simulations show that these oscillations persist for long times in agreement with the results of Kuo and Polvani. In a similar manner to what is seen in Lelong and Sundermeyer, the case with $Ro = 1$ ($f_0$ and $\frac{1}{2}w_0$) appears to retain the maximum amount of potential energy (as opposed to kinetic energy for Lelong and Sundermeyer). However, we have verified that the results seen

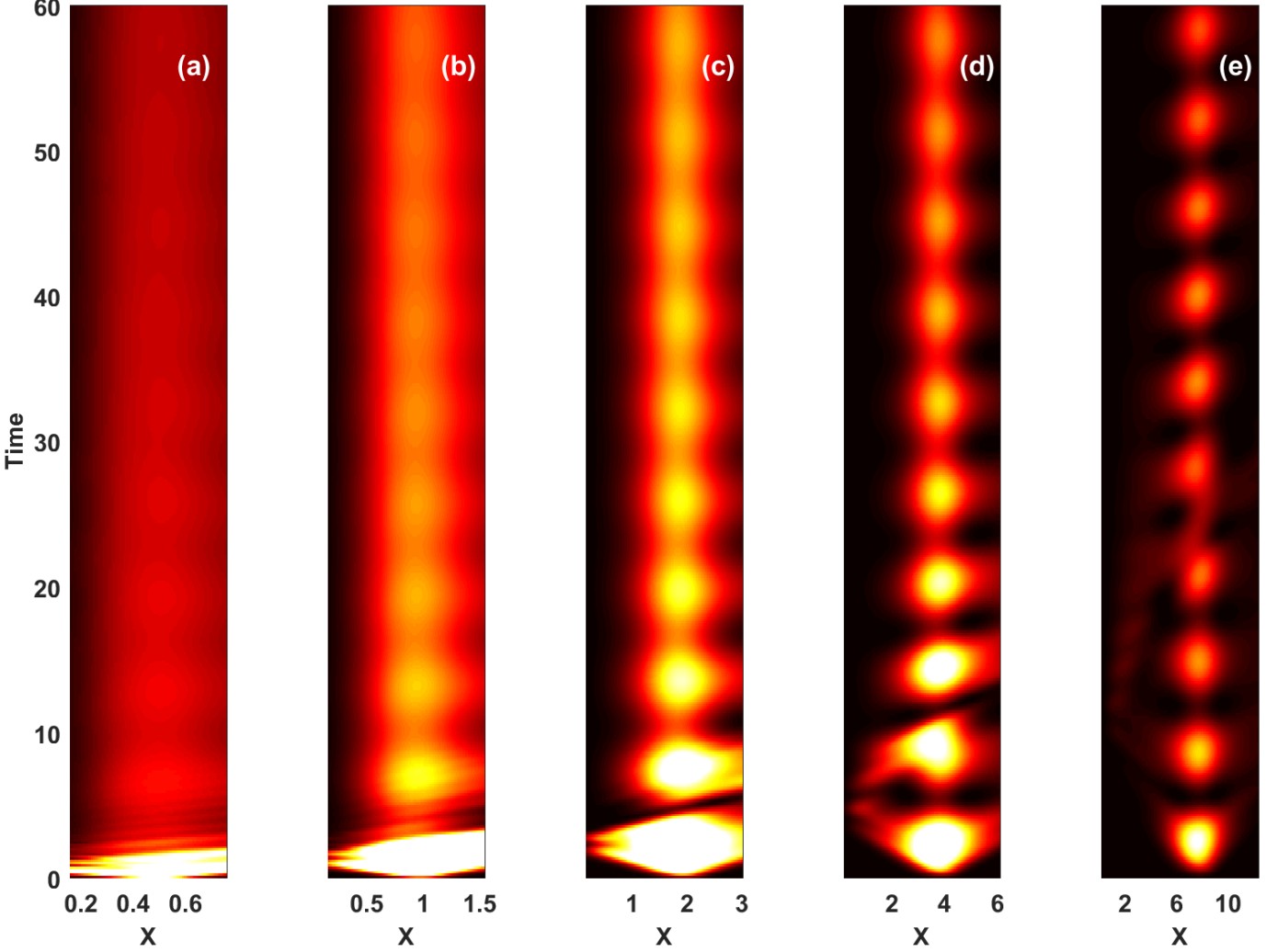

**Figure 10.** A space-time plot of vertically integrated kinetic energy, in the geostrophic state, for different values of $w$ while $f$ is held constant at $f = f_0$. (a) $w = \frac{1}{4}w_0$ ($Ro = 2$), (b) $w = \frac{1}{2}w_0$ ($Ro = 1$), (c) $w = w_0$ ($Ro = 0.5$), (d) $w = 2w_0$ ($Ro = 0.25$), and (e) $w = 4w_0$ ($Ro = 0.125$).

in their figure can be generated by scaling the kinetic energy by the initial energy (not shown here). We also computed the linear kinetic energy for the geostrophic state following Boss and Thompson (1995) equation (9), using the parameter set for our base case. We then compared this to the maximum kinetic energy in the geostrophic state. We calculated the linear KE to be $4.06327 \cdot 10^{-5}$, while our KE was $3.63771 \cdot 10^{-5}$ which is roughly an 11% difference.

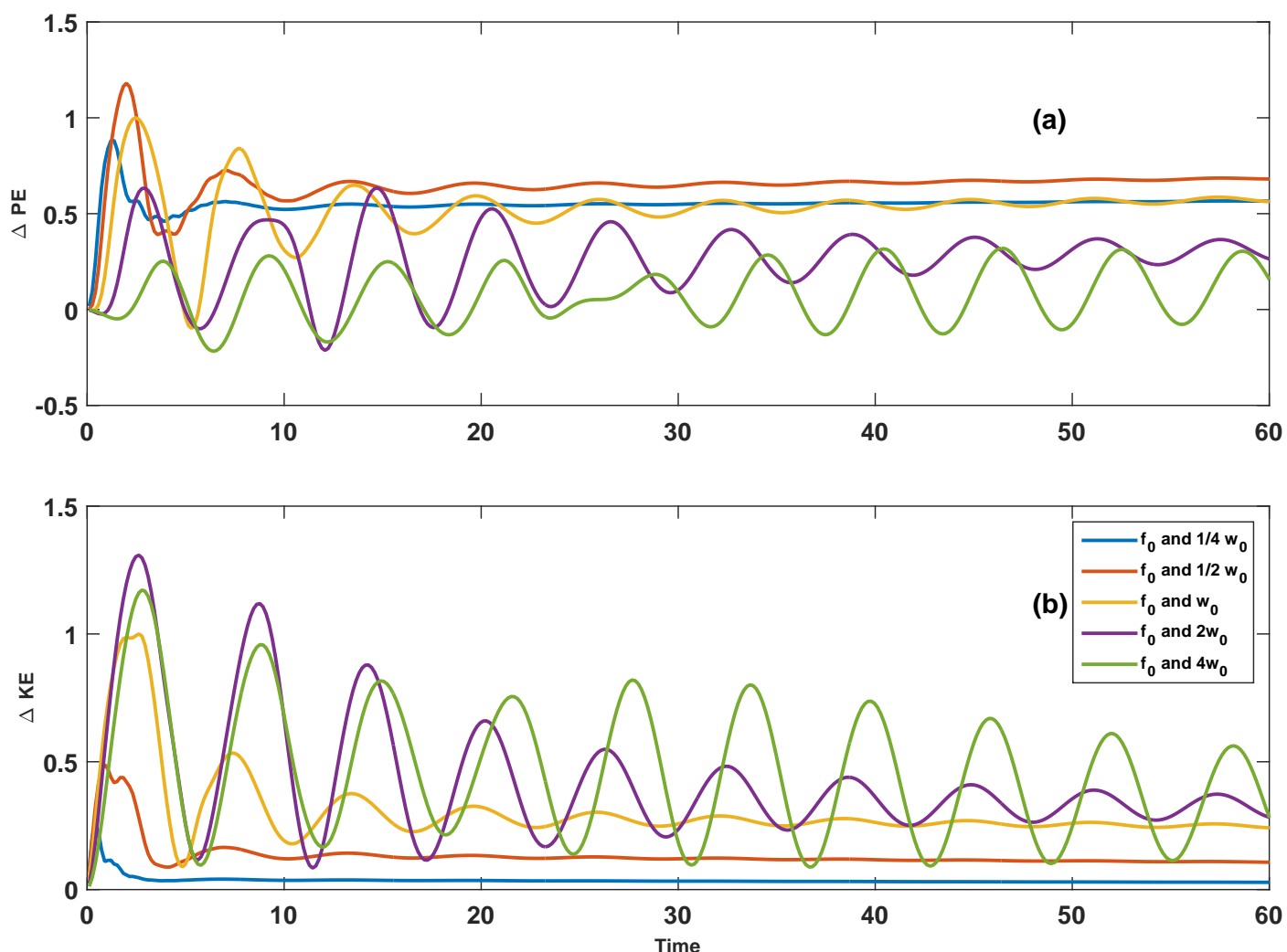

**Figure 11.** The changes in potential and kinetic energy, compared to the initialization, for the cases in Figure 6. (a) corresponds to the change in potential energy, while (b) to the changes in kinetic energy. The energy in both plots have been scaled by the base case ($f = f_0$, $w = w_0$).

We next consider how the time evolution of the total kinetic energy inside the geostrophic state compares to that outside, this is shown in Fig. 12. In this figure we have the same cases as in Fig. 10 and Fig. 11, separated into their own panels. Fig. 12 (a) corresponds to $\frac{1}{4}w_0$ ($Ro = 2$), Fig. 12 (b) corresponds to $\frac{1}{2}w_0$ ($Ro = 1$), Fig. 12 (c) corresponds to $w_0$ ($Ro = 0.5$), Fig. 12 (d) corresponds to $2w_0$ ($Ro = 0.25$) and Fig. 12 (e) corresponds to $4w_0$ ($Ro = 0.125$). The red line (inner) is the energy inside the geostrophic state, while the blue line (outer) corresponds to that outside. From Fig. 9(a) we can see that for low rotation

rates there is much less energy retained within the geostrophic state, and that the oscillations of this remaining kinetic energy are very small. We can compare this panel with figure 13 of Lelong and Sundermeyer and see similar results, namely the dominance of the kinetic energy outside the geostrophic state. We should also note that there is a steady decrease in kinetic energy in the outer kinetic energy which is due to dissipation of the waves throughout the numerical domain. As in figure 6 of Lelong and Sundermeyer we see a similar separation between the energy of the inner and outer regions for the $Ro = 1$ case, our Fig. 12 (b). The inner kinetic energy oscillations of Fig. 12 (d) and (e) do not appear to have reached a steady state by the end of our simulation, but have reached a sufficiently close value to interpret. In Fig. 12 (d) there is an equal amount of energy in the final inner and outer regions. These results seen in Fig. 12 (d) and (e) are consistent with the results from Lelong and Sundermeyer's figure 14, namely significantly more energy being retained inside the geostrophic state resulting in much larger amplitude oscillations.

Motivated by the results shown in 12 we considered the vertical structure of the inertial oscillations. We confirmed that for all cases the gradient Richardson number (including the $v$ component of shear) never dips below $0.25$ in the stratified region, thereby suggesting the inertial waves are not strong enough to induce shear instability. Indeed, the isopycnal displacements associated with the inertial oscillations were never larger than about $2.5\%$ of the total depth. For early times and low Rossby numbers ($Ro \leq 0.125$), the spatiotemporal (in $z$ and $t$) structure of the kinetic energy field induced by the inertial waves followed a separable structure. For the $Ro = 0.25$ case evidence of a nonseparable structure was clear for $t > 100$. In comparison, for the $Ro = 0.5$ case non-separable structure was evident by $t = 60$. However, since the inertial oscillations are smaller in this case at later times ($t > 150$) the signature of the inertial waves is masked by that of the geostrophic state.

## 4  Conclusions

In this paper we have taken a systematic approach to the classical rotation-modified stratified adjustment problem. Building on results based on shallow water theory presented in Kuo and Polvani (1997), we have shown that by using the fully nonlinear incompressible Navier-Stokes equations, under the Boussinesq approximation, the waves that are ejected from the geostrophic state do not steepen to a shock. Once the wave front steepens sufficiently it disperses into a primary wave packet and a tail of smaller dispersive waves. We demonstrated that the nonlinear wave packet interpretation of the wave train of Grimshaw et al. (2012) is appropriate in some parameter regimes, with changes in amplitude reflected in the phase of the nearly solitary wave response. By mapping out the parameter space we have shown that, as expected, the Rossby number is the controlling variable for the dynamics in this problem. For $Ro < 1$, the wave packet propagates with a speed roughly corresponding to the linear group speed, while for $Ro > 1$, the packet propagates with a speed closer to the linear phase speed. We have further characterized the nonlinear effects present in both the wave packet, and the geostrophic state. The effects of nonlinearity were investigated by considering different initial amplitudes, and changes in polarity. As in the non-rotating case, the largest nonlinear effects occurred as a result of changes in polarity, both in the geostrophic state and in the wave packet ejected. Surprisingly, the high Rossby number cases yielded nonlinear effects in both the wave train and the geostrophic state. However, as a general rule amplitude effects were smaller than polarity effects.

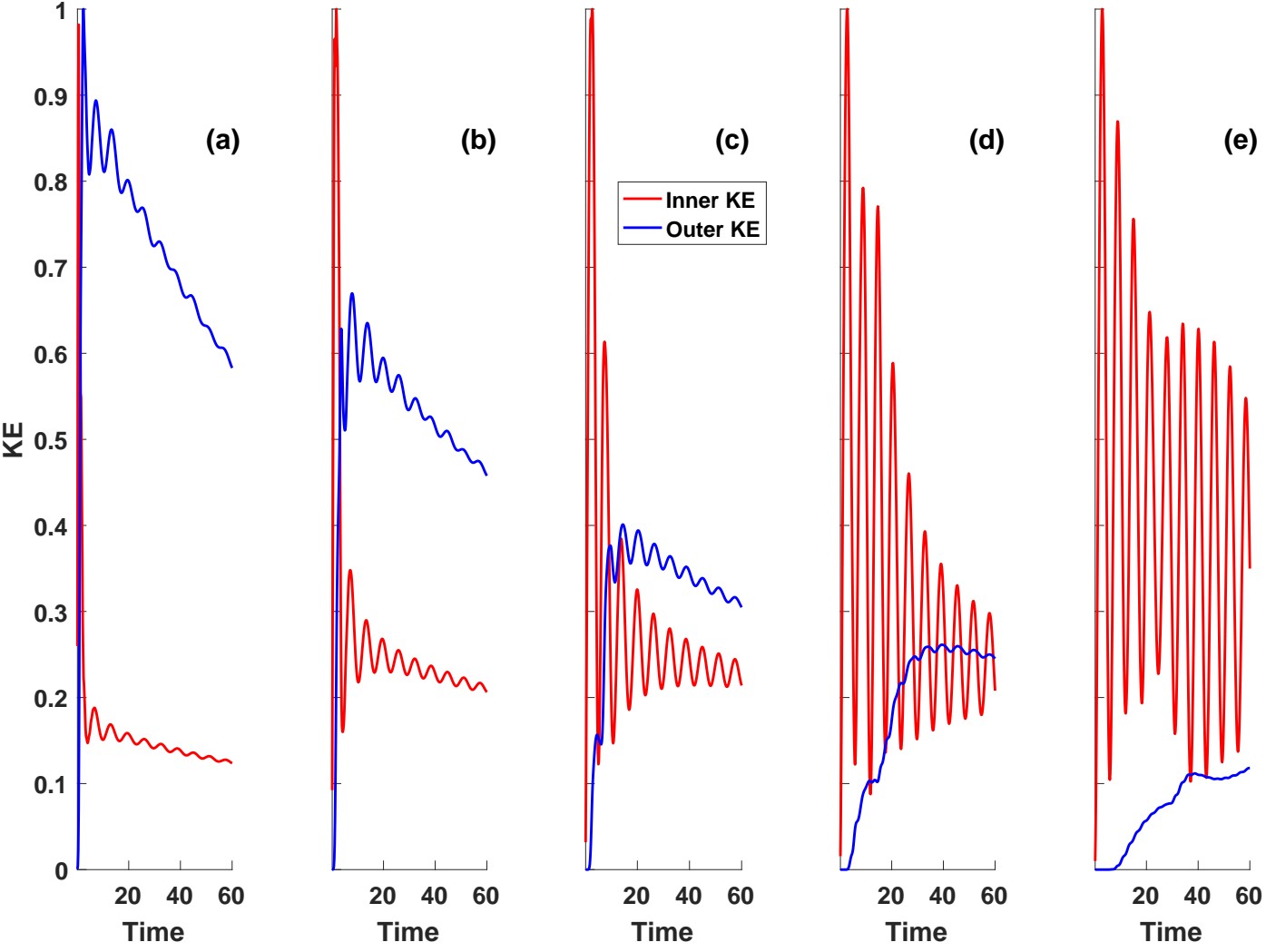

**Figure 12.** The total kinetic energy located inside (blue) and outside (red) of the geostrophic state. The different panels correspond to the different cases seen in Fig. 10: (a) $w = \frac{1}{4}w_0$ ($Ro = 2$), (b) $w = \frac{1}{2}w_0$ ($Ro = 1$), (c) $w = w_0$ ($Ro = 0.5$), (d) $w = 2w_0$ ($Ro = 0.25$), and (e) $w = 4w_0$ ($Ro = 0.125$).

A different approach to characterize nonlinear effects is to create different combinations of parameters that yield the same Rossby number. We carried out this process and tracked the time dependent Froude number. While the qualitative features of the evolution were similar in all three cases shown, the variations in the Froude number lead to significant differences in the

details of the wave train generated. The characterization of these various nonlinear effects in a single simulation is new and significant, providing a guideline for when linear theory can be applied, and when nonlinear effects must be considered.

Our results show that the inertial oscillations in the geostrophic state can persist for long times, in agreement with Kuo and Polvani (1997). However, the inertial oscillations never reach large ebough amplitudes to induce shear instability. Our results also match the work published by Lelong and Sundermeyer (2005), specifically matching their results for the different Rossby number regions ($Ro < 1$, $Ro = 1$, and $Ro > 1$). By comparing the kinetic energy within the inner geostrophic, and outer non-geostrophic regions, we show that the amplitude of the geostrophic oscillations increases quickly as Rossby number decreases. However this in turn corresponds to less prominent nonlinear effects within the geostrophic state.

Another significant finding presented is the generation, reflection and interaction of a wave train propagating in the opposite direction (leftward) during the initial generation. For any physical tank set up, this reflected wave will impact any measurements of the waves generated and especially any measurements of the geostrophic state.

While the main focus of this work is to build on the nonlinear wave literature (for which the assumption of a constant buoyancy frequency is a pathological limit), it is worth noting that geostrophic adjustment has been considered via a very different technique in the literature. Sundermeyer and Lelong (2005) performed three-dimensional simulations of a triply-periodic, linearly stratified domain. Energy was injected into the system via local patches of diffusivity, and the resulting state was allowed to adjust. The authors found that vortex–vortex interactions drive strong nonlinearity, albeit after a period of adjustment that is many inertial periods long. The detailed dynamics of the adjustment period is not considered in detail, and in any event the physical scales of the study are chosen so that our model tank would span less than 5 grid points, and hence detailed comparisons would be premature. However, an extension of the present work to the collapse of an initially mixed region and the generation of higher mode waves could provide an interesting bridge between these two presently divergent research directions.

In addition to the work described in the previous paragraph, future work should consider spanwise variations, especially in the case of the strong geostrophic state for which novel instabilities may be possible (though as noted above, on laboratory scales shear instability is not expected). Systematic studies of the shoaling of rotation-modified solitary waves and undular bores should also be carried out, since it is not known in what manner these may be different from shoaling in the non-rotating case. A more theoretical avenue could quantitatively compare weakly nonlinear and weakly dispersive-strongly nonlinear model equations to the full stratified equations.

*Acknowledgements.* This research was supported by the Natural Sciences and Engineering Research Council of Canada through Discovery Grant RGPIN 3118442010

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

**Table 1.** Rossby number of each simulation, where $f_0 = 0.105\text{s}^{-1}$ and $w_0$=1.63m.

| Ro | $\frac{1}{4}w_0$ | $\frac{1}{2}w_0$ | $w_0$ | $2w_0$ | $4w_0$ |
|---|---|---|---|---|---|
| $f_0$ | 2 | 1 | $\frac{1}{2}$ | $\frac{1}{4}$ | $\frac{1}{8}$ |
| $\frac{1}{2}f_0$ | 4 | 2 | 1 | $\frac{1}{2}$ | $\frac{1}{4}$ |
| $\frac{1}{4}f_0$ | 8 | 4 | 2 | 1 | $\frac{1}{2}$ |

**Table 2.** A comparison of the notation between our work, and that of Lelong and Sundermeyer.

| Our notation | Lelong and Sundermeyer |
|---|---|
| $Ro_r$ | $R$ |
| $Ro$ | $R/L$ |
| $w$ | $L$ |
| geostrophic state KE / PE | $KE_v$ / $PE_v$ |
| outside geostrophic state KE / PE | $KE_w$ / $PE_w$ |