# Peer review of "The Fully Nonlinear Stratified Geostrophic Adjustment Problem"

_Nonlinear Processes in Geophysics, 2015_

## Referee Comment (RC1) · Anonymous Referee #1 · 14 Mar 2016

Review of: **The Fully Nonlinear Stratified Geostrophic Adjustment Problem**

*by A. Coutino and M. Stastna*

The work presented in this manuscript is a re-examination of the mass adjustment for the classic dam-break problem with nonlinear high-resolution 2D numerical simulations. New results appear to focus on the dispersion characteristics of the radiating wavepacket and on the impact of the initial disturbance polarity. The behavior of the adjusted state for different Rossby number regimes discussed here is definitely not new. The general topic should be of interest to the general readership of the journal. However, the manuscript appears to have been submitted without careful editing (see comments below). I found the presentation of the results at times repetitious and confusing. A strong discussion rather than a mere enumeration of results would strengthen the paper. Revisions are recommended.

**Major comments:**

1. I am surprised that no mention is made of the available potential energy anywhere in this manuscript. Since it provides the initial energy source for the mass adjustment, its budget should also be considered. The available potential energy is easily calculated, as is the fraction that is converted to kinetic energy and radiated away as waves. Moreover, it would provide a nice metric for normalizing the kinetic energy and would facilitate comparison with others' results.

2. My second comment pertains to the organization of the manuscript, namely the results section. Cases where the Rossby number is varied by varying rotation rates and perturbation widths are first presented. This is then followed by another section titled "Rossby number variation". This seems somewhat repetitious. Why not present the former two as subsections of the Rossby number variation section? It would be much easier to follow the discussion of the adjusted states for the various cases considered if the Rossby numbers were given alongside the corresponding $w$ and rotation rate values. This would facilitate comparison with results from earlier studies.

3. In section 3.3, the most energetic stationary state is associated with $w = w_0$ which corresponds to $Ro = 1/2$. One might argue that values of $w$ that yield a Rossby number of $O(1)$, i.e. $w = \{0.5w_0, w_0\}$ will yield geostrophic states with the most kinetic energy. This was demonstrated in Lelong and Sundermeyer (JPO 2005) in their numerical simulations of nonlinear geostrophic adjustment. This is what Figure 5a shows though the figure is so small that it is hard to discern all the curves, i.e. where is the curve corresponding to $w = 0.25w_0$?. I assume that the KE is given as a fraction of the initial (available potential) energy though I could not find this explictily stated in the text or the figure caption. If this is the case, the fraction of KE in the adjusted state agrees with the results of Lelong and Sundermeyer (see their figure 8). Concerning the energetics of nonlinear geostrophic adjustment, another reference is Boss and Thompson (JPO 1994).

**Minor comments:**
1. A schematic of the initial condition for the 2 perturbation polarities would be helpful. This would help define the undefined parameters $H_1$, $H_2$ and $H_0$. We can guess what they represent but they nonetheless need to be defined.

2. The units used are not consistent. I would suggest sticking to mks units (i.e. do not give

some parameters in meters and others in centimeters or millimeters, e.g. on line 130 and in other places too). Also, please reconcile $\rho_0$ as defined on line 106 (dimensional) with its (nondimensiona)l value of 1 given on line 141. Units? There is also a missing factor of $\rho_0$ in the expression for the initial density expression (line 131).

3. There are numerous typos throughout the manuscript, e.g. "preformed" instead of "performed" on lines 113, 128, 134, 157 and probably elsewhere. There are also several run-on sentences (e.g. between lines 105-110, lines 88-90, lines 171-172, lines 207-209, line 325 etc.) and misspellings (e.g. "leftover" instead of "left over", "hight" instead of "height", spansws).

4. Figures are too small. They should span the width of the text. There are no colorbars on the color contour plots. Captions need to convey more information.

---

## Referee Comment (RC2) · Anonymous Referee #2 · 24 Mar 2016

**General comments**

This paper looks at the process of geostrophic adjustment in a stratified fluid, which is an important problem in atmosphere-ocean fluid dynamics. The study is based upon high-resolution fully nonlinear simulations of flow in a rotating tank at high Reynolds number, for a range of different parameters (Rossby number, width and height of initial disturbance), with the particular geometry being motivated by recent experimental results obtained at Grenoble. The nature of the ejected (nonlinear) waves and the remaining geostrophic state are discussed.

There are several good things about this article. It appears to fill in a hole in the literature for high-resolution (up to $16384 \times 192$) numerical simulations of nonhydrostatic nonlinear adjustment in 2D (which has otherwise been extensively studied analytically),

and makes a direct link to recent experimental results. The paper thus provides a useful catalogue of results, which can be used by theoreticians to test asymptotic analyses, and which supplement (imperfect or incomplete) observations from laboratory experiments. So the results are new and significant in this sense, with the numerical simulations being of an international standard. The manuscript is well-structured, of an appropriate length, with well-prepared figures. The Abstract and Introduction are well written, and would be understandable to a wide audience.

There are also several bad things about this article. In some places (e.g., section 3.2), the results are set into the context of related studies. However, this is generally not the case, particularly with regard to what extent the results can be understood or interpreted using linear theory. That is, how much of the behaviour is intrinsically nonlinear, and how much is essentially linear? I also suspect that some of the behaviour reported (e.g., the pulsations of the geostrophic state) could be understood in part using existing analytical results. The quality of the text is mediocre or poor in places, with parts reading like a PhD thesis rather than a carefully written research paper. There was also inadequate proofreading prior to submission – how can phrases such as 'unduly bore' and a 'spawns instability' (should be a spanwise instability) make it into a submitted version?

Overall, I imagine that the manuscript will be suitable for publication in NPG after moderate revisions.

**Specific comments**

1. Context.

    (a) In the Introduction, you mention the related studies of Grimshaw and Helfrich (2008) and Helfrich (2007). However, these studies are not mentioned in sections 3 or 4 (results and discussion). Is there any agreement (or disagreement) with their results? If so, this should be discussed at the relevant

points.

(b) There has been alot of other work on nonlinear geostrophic adjustment in continuously stratified fluids, albeit perhaps for slightly different systems (e.g., hydrostatic, cylindrical geometry, etc.). Comparison with some of these results would be appropriate. For example, does the theory of Zeitlin et al. (2003, JFM 491, *Nonlinear theory of geostrophic adjustment. Part 2. Two-layer and continuously stratified primitive equations*) help to understand the pulsating localised state?

(c) You say on p.3 that 'numerous papers have used a variety of methods such as asymptotic expansions and numerical integration to solve this linear problem'. So the main novelty of this paper is apparently to investigate new or different nonlinear behaviour. Some of the behaviour described is certainly nonlinear; for example, at the start of section 3.2, you discuss how a bore is replaced by a wave packet. But how much of the behaviour is essentially linear, and how much is intrinsically nonlinear? I am thinking of things like the pulsations of the localised state, the timing of the wave emissions, and the final steady geostrophic state. I think this should be made clear, by making comparisons with companion linear calculations. These could be numerical calculations, or perhaps analytical calculations for some aspects (such as the final steady state geostrophic linear solution).

Note: the only time that I see mention of varying the amplitude of the initial disturbance (second paragraph of p.15), you say that there 'was no discernible difference to any of the fields within the waves generated once they had been scaled by the factor that the amplitude was reduced', which sounds like a linear phenomenon.

2. General notation.

(a) I think it would be good practice to give the three values of $f$, $w$ and $Ro$ together, whenever possible, rather than expecting readers to continually be

turning to Table 1 (which should nevertheless be retained). For example, on line 16 of p.6 you might write '..base case with $w = w_0$ and $f = f_0$ (so $Ro = 1$), ...'; the same extra information on $Ro$ would be needed on line 22 of p.15. The three parameters values should be added to the captions of Figures 2, 5, 9, 10 and 12 (which otherwise have no information about parameters).

(b) The two different initial conditions are generally referred to as positive or negative polarity, but on two or three occasions as an initial wave of elevation or depression. I think it would be best if consistent language was used throughout.

3. p.3, lines 19,20: is there a reference for the assertion about the non-rotating adjustment?

4. p.4 (and elsewhere): in many places, references at the end of sentences should be bracketed. Examples include on lines 4, 5 and 10 of p.4. (An example of correct usage is in lines 15–16 of p.6).

5. p.7, line 12: you say that there is 'accuracy scaling with the number of grid points', but this is the case for almost all functioning numerical codes. So this statement should be removed or clarified.

6. p.7, line 22: 'the density difference was set at 1%'. This detail takes us somewhat by surprise, since the density profile has yet to be defined (other than the mention of a pycnocline on p.4).

7. pp.6–8: in (1)–(3), it seems that dimensional variables are used, except for $\overline{\rho}$ and $\rho'$, which are dimensionless according to line 9 of p.6 and line 13 of p.8. However, in the first equation on p.8, $\overline{\rho}$ takes the same dimensions as $\rho$. So presumably the equation should start with $\rho(x, z, t = 0)/\rho_0$.

8. p.8, first equation: it would be nice if $\overline{\rho}$ decreases as $z$ increases (assuming that $z$ points upwards).

9. p.8: you refer to 'the dimensionless parameter' here, but you said on p.6 that there were 'two dimensionless numbers which are dynamically important'. So maybe say 'The resulting values of $Ro$ are shown in Table 1', or 'The resulting values of this dimensionless parameter are shown in Table 1'.

10. p.8, lines 13–14: I don't believe that $H_1$ and $H_2$ have been defined, although it's easy to guess their meaning.

11. pp.7, 8: when defining $\alpha$ on line 3 of p.7, you say that '$H_0$ is the height of the undisturbed fluid column'. However, at the bottom of p.8 you then take $H_0 = 0.3$ m, rather than the tank depth of 0.4 m. What is happening here?

12. p.9 onwards: presumably $x = 0$ corresponds to the left-hand wall?

13. p.10, paragraph 2: how closely do the observed structures resemble the modulated wave packet of Grimshaw et al.? Is this a rather superficial resemblance, or are there explicit predictions to compare against?

14. p.11, lines 17–18: you say that there is 'evidence of waves being continuously ejected from the geostrophic state, even for late times in the simulation'. I can't see this evidence in Figures 3a,b, where little wave ejection is visible after about $t = 15$.

15. p.12, lines 1–2: it's almost impossible to see the generation from the fourth pulse in Fgure 4e.

16. p.28, caption to Figure 5: presumably the colours represent different initial values of $w_0$ (or Rossby numbers), rather than different 'initial rotation rates' (whatever they may be)?

17. Section 3.3: this section was lazily written.

   (a) You say that the geostrophic state is 'shown as the blue dashed line in Fig. 4'. What does this mean? Is it the centre of the geostrophic state (on the basis of kinetic energy)? Is it determined by eye, or is there an algorithm? How does the geostrophic state appear in the height field?

   (b) I think that Fig. 4 may well be better without the blue dashed line, which obscures the geostrophic state. Could it be replaced by an arrow or a dash at the top of each panel?

   (c) The kinetic energy plots are a bit confusing. The text (p.12, line 20) talks about proportions, but the numbers shown in the plots certainly don't everywhere add up to 1. Presumably these are just raw kinetic energies, in which case wouldn't it be better to show proportions, or use some other normalisation (particularly in panel (a))?

   (d) On a related point, why are we just looking at kinetic energy, rather than total energy? It seems as if you are trying to get a measure of the size of the geostrophic state, in which case wouldn't it make more sense to calculate total energy? This might mask the oscillations in figures 6c–e, but it might make a cleaner conclusion.

   (e) As noted in point 1(c), the whole discussion in this section should be grounded by a discussion of what happens in the linear case, when presumably some things are known (either for single-layer shallow water, or perhaps modes in a continuously stratified system). For example, is the drift in the centering of the geostrophic state as $w$ changes understood linearly? What is known about the K.E. (and P.E.) for the linear case?

18. p.13, line 16: 'We can also see that the oscillations of the geostrophic state remain quite evident for later times'. Really? Maybe this is visible in figure 7b,

not not in figure 7a (no oscillations visible after $t = 20$) or figure 7c (only one oscillation shown).

19. p.13, line 24: why is $Ro = 1$ picked out as being a 'critical Rossby number'? There doesn't appear to be any kind of sharp transition in the dynamics there. This is picked up again on p.14 (lines 8–10), where it is said there is a 'change in the dynamics'. However, this looks like part of a gradual change from higher Rossby number to small Rossby number, and there seems to be no basis for singling out $Ro = 1$ as a critical number.

20. p.14, 15: you make the point that the results here are fundamentally different to those based on KdV. However, this is entirely to be expected since, as far as I understand, the KdV analysis is for the non-rotating problem. I think this issue could be clarified.

21. p.15, lines 6–14: this entire paragraph seems to be saying that the observed be-haviour is almost linear. The degree of nonlinearity should be discussed explicitly. What is new here?

    As you move on to p.16 and talk about differences between positive polarity and negative polarity solutions, the results are clearly nonlinear.

22. pp.15, 16: you set up the idea that the results agree with those of Stastna et al. rather than Grimshaw et al. So please add a comment as to what is missing in the work of Grimshaw et al. For example, is that for a different asymptotic regime or physical configuration?

23. p.16, line 20: why is it clear that the 'positive polarity case is strongly affected by the boundaries'?

24. Figure 12: could you add the predictions of linear theory to this graph? It would be nice to have a quantification of the importance of nonlinearity in each case.

**Technical corrections**

1. There are many issues with punctuation in this article – particularly missing or incorrect usage of commas and semicolons. Here are some examples:

   (a) p.3, line 25: incorrect usage of a semi-colon.

   (b) p.5, line 13: missing punctuation between 'our domain' and 'this will'.

   (c) p.9, line 25: missing punctuation between 'discussion here' and 'however this'.

   (d) p.11, lines 12–14: both missing punctuation and a spelling mistake in this sentence!

   (e) p.11, line 29: should be colon (not a comma) between 'dominant features' and 'the geostrophic state'.

   (f) p.12, line 21: another example of poor punctuation.

   (g) p.13, lines 20-22: another sentence with particularly poor punctuation. The second comma (after ejected) should be something else, and the semi-colon should be a comma.

   I am stopping pointing out errors with commas and semi-colons from hereon!

2. Final line of abstract: repeated 'allows for'.

3. p.3, line 6: why are the references out of order?

4. p.3, line 22: dependant.

5. p.4, lines 13–17: this long sentence should be reworded.

6. p.5, line 19: spurious 'though'? Otherwise it's hard to make sense of this sentence.

7. On several occasions, 'performed' is written as 'preformed' (p.6 line 18, p.7 line 9, p.9 line 7).

8. p.8, line 17: presumably you mean $\times 10^{-6}$ rather than $e^{-6}$ (too much time spent on a computer).

9. p.9, line 10: hight.

10. I suspect that 'rank ordered' should always be hyphenated, as it is on p.4 but not elsewhere.

11. caption to figure 7: 'the second row corresponds to the same case as the first columns but the aspect ration has been scaled...': presumably 'first row' and 'aspect ratio'?

12. p.14, line 20: repeated 'is the'.

13. p.15, lines 13–14: how can the authors not spot 'spawns variation' and 'spawns instability', which presumably should be 'spanwise variation' and 'spanwise instability'?

14. p.15, line 7: 'From the cases that were run there was no....' could be shortened to 'There was no...'.

15. p.15, lines 23–25: it isn't necessary to repeat the information about the changed resolution (16384 and 1.59cm), which was given before.

16. p.16, line 7: 'results ... is' should be 'results ... are'.

17. p.17, line 10: 'cases exhibits'?

18. p.17, line 16: '..examined at the differences..' should be '..examined the differences..'.

19. p.17, line 26: 'unduly bore' should be 'undular bore'.

20. p.17, line 19: 'corresponds an initial wave' should be 'corresponds to an initial wave'.

---

## Author Comment (AC1) · 6 May 2016

**Response to reviewers**

We would like to thank both reviewers for their insightful comments and for their time. There were many editorial and typographic errors in the original manuscript which the reviewers pointed out and have been fixed, in the future closer attention to detail before submission will be paid. Reviewer 1's comment asking us to consider potential energy has helped strengthen the results in the Geostrophic State section and helped show the effect that the initially leftward propagating wave has on the geostrophic state. This new result is extremely relevant to physical experiments and would not have been noticed without this comment. Reviewer 2's comments about the nonlinear effects helped us develop a new section devoted to these effects. The new results indicate significant nonlinear effects in both the geostrophic state and the ejected waves for moderate to high Rossby numbers. They also provide a clearer connection to traditional nonlinear wave theory. We would like to note that significant changes to both the structure and details of the manuscript have been made and as such listing every change would be extremely difficult. In the response to each reviewer comment we have tried to indicate where the appropriate change has been made in the new manuscript. All of the comments provided were greatly appreciated and addressing them has significantly strengthened the manuscript.

**1   Reviewer 1**

**1.1   Major Comments**

1. **I am surprised that no mention is made of the available potential energy anywhere in this manuscript. Since it provides the initial energy source for the mass adjustment, its budget should also be considered. The available potential energy is easily calculated, as is the fraction that is converted to kinetic energy and radiated away as waves. Moreover, it would provide a nice metric for normalizing the kinetic energy and would facilitate comparison with others' results.**
   We computed the available potential energy for the various cases considered using both a sorted and far upstream density profile, with both choices giving essentially identical results. To more closely match the results from Kuo and Polvani, we elected to calculate the difference in total potential energy from the initial state. The potential energy was calculated for the 'primary' cases provided in Table 1. For many of the cases the potential energy did not provide any additional information compared to the kinetic energy in either the geostrophic state or the ejected waves, however this was not the case for low Rossby number simulations. In these cases the kinetic energy in the geostrophic state is dominated by the spanwise velocities. This drowns out any other structure that appears, and hence the potential energy provides more information about

the geostrophic state in these cases. A potential energy space-time plot was especially useful for identifying the leftward propagating wave. We have elected to replace Figure 5 with the corresponding $\Delta PE$ and $\Delta KE$ and mention in the text that all the cases oscillate at the inertial frequency. We have elected to normalize all the kinetic energies by the maximum of the corresponding case (or a relevant case for the figure) so as to highlight the relative differences and similarities between the cases.

2. **My second comment pertains to the organization of the manuscript, namely the results section. Cases where the Rossby number is varied by varying rotation rates and perturbation widths are first presented. This is then followed by another section titled 'Rossby number variation'. This seems somewhat repetitious. Why not present the former two as subsections of the Rossby number variation section? It would be much easier to follow the discussion of the adjusted states for the various cases considered if the Rossby numbers were given alongside the corresponding w and rotation rate values. This would facilitate comparison with results from earlier studies.**

   The section naming convention and section order has been changed to improve the organization of the paper. Under the heading results there are now subsections: 3.1 Non-rotating case, 3.2 General evolution, 3.3 The geostrophic state, 3.4 Rossby number transition and 3.5 Nonlinear and polarity effects.

3. **In section 3.3, the most energetic stationary state is associated with w = w0 which corresponds to Ro = 1=2. One might argue that values of w that yield a Rossby number of O(1), i.e. $w = 0.5w_0, w_0$ will yield geostrophic states with the most kinetic energy. This was demonstrated in Lelong and Sundermeyer (JPO 2005) in their numerical simulations of nonlinear geostrophic adjustment. This is what Figure 5a shows though the Figure is so small that it is hard to discern all the curves, i.e. where is the curve corresponding to $w = 0.25w_0$?. I assume that the KE is given as a fraction of the initial (available potential) energy though I could not find this explicitly stated in the text or the Figure caption. If this is the case, the fraction of KE in the adjusted state agrees with the results of Lelong and Sundermeyer (see their Figure 8). Concerning the energetics of nonlinear geostrophic adjustment, another reference is Boss and Thompson (JPO 1994).**

   A discussion of the Boss and Thompson (JPO 1994) paper has been added to the Introduction, and the results of this paper were used to inform the rewrite of the Results section. The general observation is that linear theory gets the order 1 story of energetics correct, but that nonlinear effects are clearly visible in either the wavetrain or the geostrophic (depending on Rossby number). The Lelong and Sundermeyer (JPO 2005) paper proved

tougher to connect to the present work due to differences in model set-up, and overall aim. For example, our entire tank would span 5 grid points in these authors' simulations. Moreover the constant buoyancy frequency limit is well known to be pathological for horizontally propagating internal waves (because the Dubreil Jacotin Long equation governing solitary waves in the nonrotating limit linearizes). We have discussed some of these issues in the Conclusions, and have come up with an avenue for future work that could bridge the gap between the two studies.

**1.2 Minor Comments**

1. **A schematic of the initial condition for the 2 perturbation polarities would be helpful. This would help define the undefined parameters $H_1$, $H_2$ and $H_0$. We can guess what they represent but they nonetheless need to be defined.**
   A schematic figure has been added to the Methods section to clearly illustrate the definition of the various parameters. We thank the reviewer for the suggestion.

2. **The units used are not consistent. I would suggest sticking to mks units (i.e. do not give some parameters in meters and others in centimeters or millimeters, e.g. on line 130 and in other places too). Also, please reconcile $\rho_0$ as defined on line 106 (dimensional) with its (nondimensional) value of 1 given on line 141. Units? There is also a missing factor of $\rho_0$ in the expression for the initial density expression (line 131).**
   The units were all changed to SI. The density expression was corrected for a typo.

3. **There are numerous typos throughout the manuscript, e.g. 'preformed' instead of 'per-formed' on lines 113, 128, 134, 157 and probably elsewhere. There are also several run-on sentences (e.g. between lines 105-110, lines 88-90, lines 171-172, lines 207-209, line 325 etc.) and misspellings (e.g. 'leftover' instead of 'left over', 'hight' instead of 'height', spanws).**
   Spelling mistakes were corrected and a better job editing will be done. A particular issue was the use of auto-correct which corrected technically correct words like 'spanwise' to grammatically correct nonsense such as 'spawns'. Sometimes this occurred after our attempt to correct the sentences and we apologize for the annoyance.

4. **Figures are too small. They should span the width of the text. There are no colorbars on the color contour plots. Captions need to convey more information.**
   Figure sizes were increased in our .tex file, however during the postprocessing process the figure sizes might change. A note will be made with the resubmission to increase figure sizes in the final copy provided to

the reviewers. The colorbars were not provided since we are scaling by the maximum value of kinetic energy for the plots, and hence all panels should range from 0-1 (unless otherwise stated in the caption). The captions were expanded to provide more information.

**2 Reviewer 2**

**2.1 Specific Comments**

1. (a) **In the Introduction, you mention the related studies of Grimshaw and Helfrich (2008) and Helfrich (2007). However, these studies are not mentioned in sections 3 or 4 (results and discussion). Is there any agreement (or disagreement) with their results? If so, this should be discussed at the relevant points.**

    The agreement with these authors' results is predominantly qualitative, since their description of the ejected waves as a localized wave-packet works as a good descriptor for the dynamics that are observed in the ejected wave train, certainly much better than the results based on shallow water theory. A mention of this has been added to the conclusions. A detailed comparison between weakly nonlinear, weakly dispersive-strongly nonlinear, and the full stratified equations is a worthwhile, but independent project.

    (b) **There has been a lot of other work on nonlinear geostrophic adjustment in continuously stratified fluids, albeit perhaps for slightly different systems (e.g., hydrostatic, cylindrical geometry, etc.). Comparison with some of these results would be appropriate. For example, does the theory of *Zeitlin et al. (2003, JFM 491, Nonlinear theory of geostrophic adjustment. Part 2. Two-layer and continuously stratified primitive equations)* help to understand the pulsating localised state?**

    We have elected to neglect linearly stratified fluids because we are concentrating on a literature that culminates in Grimshaw and Helfrich 2013, and this literature considers quasi-two layer and two layer fluids. Moreover, linear stratification is highly specialized in the sense that the Dubreil-Jacotin-Long (DJL) equation linearizes in this case and solitary wave solutions are possible. This is problematic in the high Rossby wave limit since the theoretical results on the Ostrovsky equation and the related literature generally build on the solitary wave solution and attempt to ascertain such waves' fate in the presence of rotation. A discussion of the Zeitlin et al papers, which are primarily concerned with multiple scale asymptotic analysis and hence do not directly inform the present simulations, has been added to the introduction.

(c) **You say on p.3 that 'numerous papers have used a variety of methods such as asymptotic expansions and numerical integration to solve this linear problem'. So the main novelty of this paper is apparently to investigate new or different nonlinear behaviour. Some of the behaviour described is certainly nonlinear; for example, at the start of section 3.2, you discuss how a bore is replaced by a wave packet. But how much of the behaviour is essentially linear, and how much is intrinsically nonlinear? I am thinking of things like the pulsations of the localised state, the timing of the wave emissions, and the final steady geostrophic state. I think this should be made clear, by making comparisons with companion linear calculations. These could be numerical calculations, or perhaps analytical calculations for some aspects (such as the final steady state geostrophic linear solution).**

A new section devoted to investigating the non-linearity of the simulations was added. The appropriately scaled horizontal profiles of vertically integrated KE of several different rotation modified cases were compared against the non-rotating case. Were the results to be linear all these profiles should collapse to the same curve. This was found not to be the case, with polarity changes yielding the greatest difference. These results show that there are fairly sizeable non-linear effects in both the geostrophic state (for low Rossby number) and in the ejected waves (for high Rossby number). For lower Rossby number the energy in the ejected wave train is less and hence linearity is a better approximation . Similarly, for low rotation rates, the geostrophic state decreases in amplitude and hence is better approximated by linear theory.

2. (a) **I think it would be good practice to give the three values of $f$, $w$ and Ro together, whenever possible, rather than expecting readers to continually be turning to Table 1 (which should nevertheless be retained). For example, on line 16 of p.6 you might write ..base case with $w = w_0$ and $f = f_0$ (so Ro = 1), ...; the same extra information on Ro would be needed on line 22 of p.15. The three parameters values should be added to the captions of Figures 2, 5, 9, 10 and 12 (which otherwise have no information about parameters).**

All three values have been added to all the captions and in text.

(b) **The two different initial conditions are generally referred to as positive or negative polarity, but on two or three occasions as an initial wave of elevation or depression. I think it would be best if consistent language was used throughout.**

The language has been made consistent.

3. **p.3, lines 19,20: is there a reference for the assertion about the non-rotating adjustment?**
A new discussion summarizing the present understanding of this case has been added to the Results section.

4. **p.4 (and elsewhere): in many places, references at the end of sentences should be bracketed. Examples include on lines 4, 5 and 10 of p.4. (An example of correct usage is in lines 1516 of p.6).**
The references have been fixed.

5. **p.7, line 12: you say that there is 'accuracy scaling with the number of grid points', but this is the case for almost all functioning numerical codes. So this statement should be removed or clarified.**
The sentence has been changed to 'order of accuracy', and this corrected is assertion is only true for spectral methods.

6. **p.7, line 22: 'the density difference was set at 1% '. This detail takes us somewhat by surprise, since the density profile has yet to be defined (other than the mention of a pycnocline on p.4).**
The density difference of 1% was chosen to match the experiments of Grimshaw and Helfrich 2013, a sentence has been added to clarify this.

7. **pp.68: in (1)(3), it seems that dimensional variables are used, except for $\bar{\rho}$ and $\rho'$, which are dimensionless according to line 9 of p.6 and line 13 of p.8. However, in the first equation on p.8, $\bar{\rho}$ takes the same dimensions as $\rho$. So presumably the equation should start with $\rho(x; z; t = 0)/\rho_0$.**
There was a typo in the density equations, it has since been corrected.

8. **p.8, first equation: it would be nice if $\bar{\rho}$ decreases as z increases (assuming that z points upwards).**
There was a typo in the density equations, it has been corrected and a stable profile results.

9. **p.8: you refer to 'the dimensionless parameter' here, but you said on p.6 that there were 'two dimensionless numbers which are dynamically important'. So maybe say 'The resulting values of Ro are shown in Table 1', or 'The resulting values of this dimensionless parameter are shown in Table 1'.**
This sentence has been changed.

10. **p.8, lines 1314: I don't believe that $H_1$ and $H_2$ have been defined, although it's easy to guess their meaning.**
A schematic figure has been added which should provide a clear pictorial definition of these parameters.

11. **pp.7, 8: when defining $\alpha$ on line 3 of p.7, you say that '$H_0$ is the height of the undisturbed fluid column'. However, at the bottom of p.8 you then take $H_0 = 0.3$ m, rather than the tank depth of 0.4 m. What is happening here?**
A schematic figure has been added which should provide a clear pictorial definition of these parameters.

12. **p.9 onwards: presumably $x = 0$ corresponds to the left-hand wall?**
A sentence has been added to clarify this.

13. **p.10, paragraph 2: how closely do the observed structures resemble the modulated wave packet of Grimshaw et al.? Is this a rather superficial resemblance, or are there explicit predictions to compare against?**
The ejected waves quite closely resemble the modulated wave-packet of Grimshaw et al., however in contrast to their predictions, even at late times the packet does not separate from the tail. In general it is not possible to directly quantitatively compare with their results since they do not model the geostrophic state, and since their theory neglects the leftward propagating wave which reflects off the left wall and influences the dynamics. For these reasons we leave our comparison with their results to a qualitative one.

14. **p.11, lines 1718: you say that there is 'evidence of waves being continuously ejected from the geostrophic state, even for late times in the simulation'. I can't see this evidence in Figures 3a,b, where little wave ejection is visible after about t = 15.** The line has been removed.

15. **p.12, lines 12: it's almost impossible to see the generation from the fourth pulse in Fgure 4e.**
The figures have been increased in size in the manuscript and a note will be made for resubmission.

16. **p.28, caption to Figure 5: presumably the colours represent different initial values of $w_0$ (or Rossby numbers), rather than different 'initial rotation rates' (whatever they may be)?**
This figure has been replaced with a figure of KE and PE, the caption has been corrected.

17. (a) **You say that the geostrophic state is 'shown as the blue dashed line in Fig. 4'. What does this mean? Is it the centre of the geostrophic state (on the basis of kinetic energy)? Is it determined by eye, or is there an algorithm? How does the geostrophic state appear in the height field?**
We have replaced the following figure with one showing PE and KE

within the geostrophic state which as been defined as twice the distance from the peak in kinetic energy to the left-hand wall.

(b) **I think that Fig. 4 may well be better without the blue dashed line, which obscures the geostrophic state. Could it be replaced by an arrow or a dash at the top of each panel?**
The line has been removed since we no longer use the figure that it refers to.

(c) **The kinetic energy plots are a bit confusing. The text (p.12, line 20) talks about proportions, but the numbers shown in the plots certainly don't everywhere add up to 1. Presumably these are just raw kinetic energies, in which case wouldn't it be better to show proportions, or use some other normalisation (particularly in panel (a))?**
Unless otherwise stated in the caption, the kinetic energies are scaled by their maximum so as to highlight the geometrical structure between cases. Figure 5 was scaled by the maximum of all the scales to show the relative energy in each case.

(d) **On a related point, why are we just looking at kinetic energy, rather than total energy? It seems as if you are trying to get a measure of the size of the geostrophic state, in which case wouldn't it make more sense to calculate total energy? This might mask the oscillations in figures 6ce, but it might make a cleaner conclusion.**
We have replaced this Figure with a new figure that shows both the change in kinetic energy and the potential energy. These were calculated separately to relate to the commonly used ratio $\Delta KE/\Delta PE$ (though we cannot calculate it directly in this case). In general kinetic energy is used as it provides both information about the structure and velocity.

(e) **As noted in point 1(c), the whole discussion in this section should be grounded by a discussion of what happens in the linear case, when presumably some things are known (either for single-layer shallow water, or perhaps modes in a continuously stratified system). For example, is the drift in the centering of the geostrophic state as $w$ changes understood linearly? What is known about the K.E. (and P.E.) for the linear case?**
As mentioned in the response to point 1(c) we have devoted a new section to the effects of nonlinearity and have considered a 'nearly linear' case. Furthermore for our smooth initial conditions the peak in kinetic energy always occurs where the isopycnals transition. Thus they should scale linearly with disturbance width. In regards to the energetics, according to Boss and Thompson, 1994 the linear estimate of the kinetic energy should be a very good approximation to the nonlinear case.

18. **p.13, line 16: 'We can also see that the oscillations of the geostrophic state remain quite evident for later times'. Really? Maybe this is visible in figure 7b, not in figure 7a (no oscillations visible after $t = 20$) or figure 7c (only one oscillation shown).**
    The figures have been enlarged to help identify this feature, and the new figures 7 and 9 clearly show the oscillations.

19. **p.13, line 24: why is Ro = 1 picked out as being a 'critical Rossby number'? There doesn't appear to be any kind of sharp transition in the dynamics there. This is picked up again on p.14 (lines 810), where it is said there is a 'change in the dynamics'. However, this looks like part of a gradual change from higher Rossby number to small Rossby number, and there seems to be no basis for singling out Ro = 1 as a critical number.**
    Though there may not be a discrete change across Ro=1, there is a shift in the dynamics, as highlighted in Figure 10. If we classify the location of the ejection of the wave packet as a metric for this transition, there is a shift when we pass Ro=1. The reviewer's point is well taken, though, and we have worked to temper the language around this point.

20. **p.14, 15: you make the point that the results here are fundamentally different to those based on KdV. However, this is entirely to be expected since, as far as I understand, the KdV analysis is for the non-rotating problem. I think this issue could be clarified.**
    In the non-rotating case, the waves generated can be, at least qualitatively, described by KdV theory, one would then logically expect that the rotating version of this problem could be modeled by a rotation modified KdV theory (like the rKdV equation). The sentence has been changed accordingly.

21. **p.15, lines 614: this entire paragraph seems to be saying that the observed behaviour is almost linear. The degree of nonlinearity should be discussed explicitly. What is new here? As you move on to p.16 and talk about differences between positive polarity and negative polarity solutions, the results are clearly nonlinear.**
    This paragraph has been removed and a new section devoted to nonlinearity has been added.

22. **pp.15, 16: you set up the idea that the results agree with those of Stastna et al. rather than Grimshaw et al. So please add a comment as to what is missing in the work of Grimshaw et al. For example, is that for a different asymptotic regime or physical configuration?**
    The primary feature that is missing from the model in Grimshaw et al. is the lack of nonlinear dispersion. This could be added to the model by considering more terms in the asymptotic expansion. The reality is that

these effects are quite small and this has been stated at a number of points in the article. The model equation also neglects the geostrophic state which is inherently created during any initialization. Thus the advantage of our numerical simulations is that they capture all the effects in one convenient package. These comments have been added in the conclusion section.

23. **p.16, line 20: why is it clear that the 'positive polarity case is strongly affected by the boundaries'?**
    This line has been removed.

24. **Figure 12: could you add the predictions of linear theory to this graph? It would be nice to have a quantification of the importance of nonlinearity in each case.**

    We have added the results of a case with 1/200 of the base case's amplitude to the new Figure 11. This new figure shows quite clearly the manner in which nonlinear effects manifest themselves, and in particular the dominant effect of polarity. A more detailed discussion is provided in the new section on nonlinearity.

---

## Referee Report (RR1)

**A second review of**

**'The fully nonlinear stratified geostrophic adjustment problem',**

**by A. Coutino and M. Stastna**

**General comments**

As for the original version, this paper looks at the process of geostrophic adjustment in a stratified fluid, which is an important problem in atmosphere-ocean fluid dynamics. The study is based upon high-resolution fully nonlinear simulations of almost two-layer flow in a rotating tank at high Reynolds number, for a range of different parameters (Rossby number, width and height of initial disturbance), with the particular geometry being motivated by recent experimental results obtained at Grenoble. The nature of the ejected (nonlinear) waves and the remaining geostrophic state are discussed, along with their dependence upon the initial amplitude and polarity of the disturbance.

As for the original version, there are several good things about this article. It appears to fill in a hole in the literature for high-resolution (up to $16384 \times 192$) numerical simulations of nonhydrostatic nonlinear adjustment in 2D (which has otherwise been extensively studied analytically), and makes a direct link to recent experimental results. The paper thus provides a useful catalogue of results, which can be used by theoreticians to test asymptotic analyses, and which supplement (imperfect or incomplete) observations from laboratory experiments. So the results are new and significant in this sense, with the numerical simulations being of an international standard. The manuscript is well-structured, of an appropriate length, with well-prepared figures. The Abstract and Introduction are well written, and would be understandable to a wide audience.

This is a considerable improvement over the original version, with most of the original shortcomings rectified, and with a useful new section on nonlinearity and polarity (although the text there could be polished up). I also (still) wonder how much of the behaviour in sections 3.2–3.4 could be understood in broad terms using linear theory, since it has been shown (in section 3.5, e.g., Figures 11(b,c)) that the leading-order behaviour in certain cases is approximately linear. The paper would certainly be better and offer more insight if some simple theory (just based on comparisons with the linear dispersion relation for inertia-gravity waves) was deployed. One might argue that the main emphasis here is simply on presenting the novel numerical results and setting them in the context of previous studies, but a really good paper should also include appropriate theoretical explanations.

I think that the manuscript will be suitable for publication in NPG after some more minor revisions, mostly relating to the clarity of the text.

**Specific comments and technical corrections**

1. p.1, lines 15-22. It might be helpful to say that the 'linear problem' was first considered by Rossby, since it's not obvious until line 22 that you are indeed first talking about 'this linear problem'.

2. p.3, line 22: we don't yet know the tank geometry, so the significance of a 'leftward wave' is lost.

3. p.3, line 26: probably best to avoid the use of the word 'reflect' here – maybe reserve it for wave reflection? Perhaps '..expected to evolve according to linear theory'?

4. p.4, line 21: $f$ is not the rotation rate, but either twice the rotation rate, or the Coriolis parameter.

5. p.5, line 2: presumably the physical experiments were by Grimshaw et al. (2013), not Grimshaw and Helfrich (2008)?

6. p.5, line 3: presumably 'four times longer', rather than 'four times larger'?

7. p.5, line 6: 'the density difference was set to 1%' would belong better at the top of p.6, where the form of the initial density is first discussed.

8. p.5: presumably the physical experiments had a free surface, whereas you have a rigid lid? This could be clarified.

9. p.4, 6: on p.4 you say that $\alpha = \eta/H_1$, where $H_1$ is the height of the undisturbed fluid column – although, according to figure 1, $H_1$ is the depth of the lower layer, or height of the undisturbed interface. Then, when defining $\alpha$ on around line 13 p.6, you seem to confuse $H_0$ with $H_1$. This should all be clarified.

10. p.6, line 10: $1e^{-6}$ is a bit careless (and incorrect as written, I think).

11. p.6, line 24: is this scaled by the maximum kinetic energy (in space) at fixed $t$, or over all space and time? Is this just for certain figures?

12. p.7, line 5: The $\rightarrow$ the.

13. p.7, line 11: maybe 'dispersion coefficient $r_{01}$ ... and nonlinear coefficient $r_{10}$'?

14. p.8, start of section 3.2: should it be clear that we are now looking at negative polarity cases?

15. p.9, line 20: this sentence needs to be rewritten (unclear, and grammatically incorrect).

16. p.10, line 11: 'as we increase the *initial width*, the shape of...' would be better (so we don't have to translate 'width of the initialization' to the previously established terminology of 'initial width').

17. p.10, 11: as in the original manuscript, it's still hard to see wave emission in fig 6(e), and to a lesser extent in fig 6(d).

18. p.11, line 6: presumably the *extent* or *location* of the geostrophic region is defined as given.

19. p.11, line 7/8: 'we cannote compute $\Delta PE/\Delta KE$ since the potential energy may be zero since it can reach its initial starting point': this needs a rewrite to be clear (and correct)!

20. p.11, line 11: 'quick decay' is an odd term for something that is not tending to zero. Maybe 'rapid equilibration' would be better? Same issue on line 1 of p.12.

21. p.12, caption to Fig.8: should be 'scaled by the..'.

22. p.12, line 4: patters $\rightarrow$ patterns.

23. p.13, line 6: obvious pair of typos with superscripts to be corrected.

24. section 3.3 (and perhaps 3.2 and 3.4, too): no simple arguments are given for any of this behaviour – surely something useful could be said based on elementary theory? For example, on line p.3 of p.14, can the 'spreading of the wave packet' be interpreted in terms of the linear dispersion relation for rotating shallow-water waves (i.e., does $\partial^2 \omega/\partial k^2$ increase as $Ro$ decreases)? On p.15, you do state that the wave packet behaviour 'to leading order can be understood from the point of view of linear dispersive wave theory' – so surely you should be using this to explain the behaviour?

25. p.16, caption to Fig 10: 'above $Ro = 1.25$ case' and 'below $Ro = 0.75$ case' are misleading – better just to write $Ro = 1.25$ and $Ro = 0.75$.

26. p.14, line 4: 'crossing from below to above one Rossby number..' needs to be rewritten to be clear (and correct)!

27. The text in the new section 3.5 was hard (but not impossible) to understand, because it hadn't been written (or edited) with sufficient care. Some specific points:

    (a) p.15, line 10: 'reflecting the fact that' → 'since'?

    (b) p.16, line 7: an ambiguous sentence. 'In contrast to what was observed before' (presumably you are talking about fig 11(b)?), 'for the lower rotation case' (presumable you mean 'for this lower rotation case')?

    (c) p.16, line 15: 'Panel (a) clearly shows the energy difference that was seen in Fig.11(b)' – misleading, since it sounds like this is a different view of the same data as Fig.11(b) (which was at different parameters). You mean something like the 'same kind of energy difference between positive and negative polarity cases that was also seen in Fig.11(b).'.

    (d) caption to Figure 12: it's simply incorrect that the figure shows negative (left column) and positive (right column) – this is only in panels (c,d).

    (e) p.17, line 5: should be Figure 12(b), not 11(b).

    (f) caption to Figure 14: perhaps rewrite 'wave of elevation' and 'wave of depression' in terms of polarity (as elsewhere in text).

28. Will the URLs be removed from the final reference list?

29. caption to Table 1: obvious typo with superscript.

---

## Referee Report (RR2)

Review of: **The Fully Nonlinear Stratified Geostrophic Adjustment Problem**

*by A. Coutino and M. Stastna*

The writing and presentation has improved on this resubmission but, unfortunately, the depth of analysis has not. Results of numerical simulations of mass adjustment for the classic dam-break problem with nonlinear high-resolution 2D numerical simulations are presented, with a focus on the dispersion characteristics of the radiating wavepacket, on the impact of the initial disturbance polarity and the role of the Rossby number. The authors are applied mathematicians, yet only 4 equations appear in the paper: the equations of motion and a KdV equation. In fact, it is not clear why the latter equation is even included since it is not solved nor used at all in the manuscript. Comparisons with previous work are confined to very broad generalizations. Comparison of simulations in different regimes are equally qualitative. We are not told why Ro=1 represents a transition in behavior. What is the impact of the Reynolds number? Why not mention the Froude number? The analysis remains primarily confined to looking at kinetic energy Hövmöller plots and isopycnal displacements. Numerical simulations can be a powerful tool when used in conjunction with some theory, but here no hint of theory is presented. What have the authors learned from this study? How does energy in the geostrophic state and radiating waves depend on the Rossby number or the nonlinearity parameter? What is the role of the Reynolds number? Why include an entire table of Reynolds numbers when the impact of dissipation is not ever discussed? In my first review, I mentioned that the authors might compare their results with Lelong and Sundermeyer (JPO 2005). Instead, they concentrated on Sundermeyer and Lelong (JPO 2005) which, I agree, has no bearing on the current study. I am sorry that I cannot recommend publication of this manuscript in its present incarnation.

**Comments:**

1. Abstract, line 7: How do variations in the Rossby number *demonstrate* the presence of two wave trains?

2. Page 3, line 32: Since the set-up is on experimental scales, the *flat ocean bottom* should be *flat bottom.*

3. Page 4, As in the previous version, the definition of $\rho$ on line 7 is still not consistent with the expression for $\rho$ given on Page 6 (top of page).

4. Page 4, line 15:

5. Page 4, top of the page, equations 1-3: These are the Navier-Stokes equations, not the Euler equations which are, by definition, inviscid.

6. Page 4, line 15: Why not state Lamb's equation 14 explicitly? Nowhere in the manuscript does an energy equation appear. Which terms are dominant in the different regimes?

7. Page 7, paragraph starting on line 5: It not clear why the paragraph on the KdV theory is included since no explicit comparison with numerical results is made. You are solving the NS equations, not the KdV or DJL equation. Do your numerical results match the solutions of either of these equations? In Eq (4), what is B?

8. Page 8: Please state the polarity of your base case.

9. Page 9, lines 10-11: Those 2 sentences can be combined: Figure 4(a) corresponds to the base case (Ro=1/2), 4(b) to $f = f_0/2$ (Ro=1) and 4(c) to $f = f_0/4$ (Ro=2).

10. Page 9, Sentence starting on line 19 does not make sense.

11. Page 9, lines 28-30 run-on sentence.

12. Page 11, lines 8-9: '... *since the potential energy may be zero since it can reach its initial starting point*'. Do you mean to say that $\Delta PE$ can be 0? What's wrong with that? The difficulty in computing $\Delta PE/\Delta KE$ may come from $\Delta KE$ vanishing at $t = 0$ but typically, one is interested in the value of this ratio at large time.

13. and so on....

---

## Referee Report (RR3)

Review of: **The Fully Nonlinear Stratified Geostrophic Adjustment Problem, Version 3**

*by A. Coutino and M. Stastna*

The third iteration of this manuscript is much improved by the addition of theory delineating linear and nonlinear effects. I still caught a number of errors in the text that could have been eliminated with a final careful read-through prior to resubmission. I may not have caught all the remaining discrepancies and I suggest that the authors go over the text again very thoroughly. I recommend publication after the corrections listed below are made.

   **Comments:**

1. Page 10, line 2: In Equation 10, the factor multiplying $B_{xx}$ on the right-hand-side should be $H^2/(\pi^2 Fr^2)$ i.e. $Fr^2$ in the denominator and not $Ro^2$.

2. Page 21, line 27: it is the *change* in potential energy that may be zero, not the potential energy. This sentence needs to be rewritten for clarity.

3. Page 23, bottom of page: Sentence beginning on line 2 and ending on line 5 belongs in the caption for Figure 12.

4. Page 25, the caption of Figure 12 is wrong. Please use the text on Page 23. The colors are reversed: the kinetic energy *inside* is red and *outside* is blue.

5. Page 26, material from middle of line 14 to middle of line 19 was retained from version 2 of the manuscript and should be taken out since it is not relevant to the discussion. Also, no need to cite Sundermeyer and Lelong here.

---

## Author Response (AR2)

**Response to reviewers**

**We have revised the manuscript in its entirety, focusing on the aspects that differ from the existing literature, and in particular the nonlinear aspects of the problem. We have expanded the theoretical discussion in a number of places, including a comparison with the predictions of linear dispersive wave theory. The two sets of reviews have been divergent from the beginning, and we have endeavoured to balance the point of view we have, with those of the two reviewers. As an example, while Reviewer 2 finds no value in the discussion of the KdV equation, the discussion of the role of polarity was included as a response to an earlier comment by Reviewer 1 which (in our opinion) has improved the stand alone nature of the presentation. Detailed responses are provided below with our responses in bold.**

**Reviewer 1**

**We appreciate the many constructive suggestions the Reviewer makes and have included the results of linear theory earlier on in the discussion. We have attempted to focus the discussion on the nonlinear aspects of the behaviour (which are surprisingly varied), building on a polished (hopefully) version of the section on nonlinearity and polarity. Detailed responses can be found below (though we note that much of the text has changed).**

As for the original version, this paper looks at the process of geostrophic adjustment in a stratified fluid, which is an important problem in atmosphere-ocean fluid dynamics. The study is based upon high-resolution fully nonlinear simulations of almost two-layer flow in a rotating tank at high Reynolds number, for a range of different parameters (Rossby number, width and height of initial disturbance), with the particular geometry being motivated by recent experimental results obtained at Grenoble. The nature of the ejected (nonlinear) waves and the remaining geostrophic state are discussed, along with their dependence upon the initial amplitude and polarity of the disturbance. As for the original version, there are several good things about this article. It appears to fill in a hole in the literature for high-resolution (up to 16384 x 192) numerical simulations of nonhydrostatic nonlinear adjustment in 2D (which has otherwise been extensively studied analytically), and makes a direct link to recent experimental results. The paper thus provides a useful catalogue of results, which can be used by theoreticians to test asymptotic analyses, and which supplement (imperfect or incomplete) observations from laboratory experiments. So the results are new and significant in this sense, with the numerical simulations being of an international standard. The manuscript is well-structured, of an appropriate length, with well-prepared figures. The Abstract and Introduction are well written, and would be understandable to a wide audience. This is a considerable improvement over the original version, with most of the original shortcomings rectified, and with a useful new section on nonlinearity and polarity (although the text there could be polished up). I also (still) wonder how much of the behaviour in sections 3.23.4 could be understood in broad terms using linear theory, since it has been shown (in section 3.5, e.g., Figures 11(b,c)) that the leading-order behaviour in certain cases is approximately linear. The paper would certainly be better and offer more insight if some simple theory (just based on comparisons with the linear dispersion relation for inertia-gravity waves) was deployed. One might argue that the main emphasis here is simply on presenting the novel numerical results and setting them in the context of previous studies, but a really good paper should also include appropriate theoretical explanations. I think that the manuscript will be suitable for publication in NPG after some more minor revisions, mostly relating to the clarity of the text.

1. p.1, lines 15-22. It might be helpful to say that the linear problem was first considered by Rossby, since its not obvious until line 22 that you are indeed first talking about this linear problem.
   **This has been corrected.**

2. p.3, line 22: we dont yet know the tank geometry, so the significance of a leftward wave is lost.
   **The wording has been changed.**

3. p.3, line 26: probably best to avoid the use of the word reflect here  maybe reserve it for wave reflection? Perhaps ..expected to evolve according to linear theory?
   **The wording has been changed.**

4. p.4, line 21: f is not the rotation rate, but either twice the rotation rate, or the Coriolis parameter.
   **This has been fixed throughout the manuscript.**

5. p.5, line 2: presumably the physical experiments were by Grimshaw et al. (2013), not Grimshaw and Helfrich (2008)?
   **This has been corrected.**

6. p.5, line 3: presumably four times longer, rather than four times larger?
   **The wording has been changed.**

7. p.5, line 6: the density difference was set to 1% would belong better at the top of p.6, where the form of the initial density is first discussed.
   **This has been moved.**

8. p.5: presumably the physical experiments had a free surface, whereas you have a rigid lid? This could be clarified.
   **An explicit statement has been added.**

9. p.4, 6: on p.4 you say that , where $H_1$ is the height of the undisturbed fluid column  although, according to figure 1, $H_1$ is the depth of the lower layer, or height of the undisturbed interface. Then, when defining on around line

13 p.6, you seem to confuse $H_0$ with $H_1$. This should all be clarified.
**This has been corrected throughout the section.**

10. p.6, line 10: 1e6 is a bit careless (and incorrect as written, I think).
**This has been fixed.**

11. p.6, line 24: is this scaled by the maximum kinetic energy (in space at fixed t, or over all space and time? Is this just for certain figures?
**This has been more explicitly stated.**

12. p.7, line 5: The→the.
**This has been changed.**

13. p.7, line 11: maybe dispersion coefficient $r_{01}$ ... and nonlinear coefficient $r_{10}$?
**This has been changed.**

14. p.8, start of section 3.2: should it be clear that we are now looking at negative polarity cases?
**This has been clarified.**

15. p.9, line 20: this sentence needs to be rewritten (unclear, and grammatically incorrect).
**This has been corrected.**

16. p.10, line 11: as we increase the initial width, the shape of... would be better (so we dont have to translate width of the initialization to the previously established terminology of initial width).
**This has been corrected.**

17. p.10, 11: as in the original manuscript, its still hard to see wave emission in fig 6(e), and to a lesser extent in fig 6(d).
**This figure is no longer included.**

18. p.11, line 6: presumably the extent or location of the geostrophic region is defined as given.
**The calculation of the geostrophic region has been more explicitly stated.**

19. p.11, line 7/8: we cannot compute $\Delta PE = \Delta KE$ since the potential energy may be zero since it can reach its initial starting point: this needs a rewrite to be clear (and correct)!
**This has been corrected.**

20. p.11, line 11: quick decay is an odd term for something that is not tending to zero. Maybe rapid equilibration would be better? Same issue on line 1 of p.12.
**The wording has been changed.**

21. p.12, caption to Fig.8: should be scaled by the...
    **The wording has been changed.**

22. p.12, line 4: patters→patterns.
    **The wording has been changed.**

23. p.13, line 6: obvious pair of typos with superscripts to be corrected.
    **This has been corrected.**

24. section 3.3 (and perhaps 3.2 and 3.4, too): no simple arguments are given
    for any of this behaviour  surely something useful could be said based on
    elementary theory? For example, on line p.3 of p.14, can the spreading of
    the wave packet be interpreted in terms of the linear dispersion relation
    for rotating shallow-water waves (i.e., does $\partial^2\omega/\partial k^2$ increase as Ro de-
    creases)? On p.15, you do state that the wave packet behaviour to leading
    order can be understood from the point of view of linear dispersive wave
    theory  so surely you should be using this to explain the behaviour?
    **More linear theory has been added to frame the discussion of the
    nonlinear effects. Furthermore we have added lines indicating
    the theoretical wave front location according to linear theory to
    the new figure 3. We have also expanded the theoretical discus-
    sion to include rotating theory as well, so as to provide a link
    with the well studied Klein Gordon equation.**

25. p.16, caption to Fig 10: above Ro = 1.25 case and below Ro = 0.75 case
    are misleading  better just to write Ro = 1.25 and Ro = 0.75.
    **This section has been removed.**

26. p.14, line 4: crossing from below to above one Rossby number.. needs to
    be rewritten to be clear (and correct)!
    **This section has been removed.**

27. The text in the new section 3.5 was hard (but not impossible) to un-
    derstand, because it hadn't been written (or edited) with sufficient care.
    Some specific points:
    (a) p.15, line 10: reflecting the fact that→since?
    (b) p.16, line 7: an ambiguous sentence. In contrast to what was observed
    before (presumably you are talking about fig 11(b)?), for the lower rota-
    tion case (presumable you mean for this lower rotation case)?
    (c) p.16, line 15: Panel (a) clearly shows the energy difference that was
    seen in Fig.11(b)  misleading, since it sounds like this is a different view
    of the same data as Fig.11(b) (which was at different parameters). You
    mean something like the same kind of energy difference between positive
    and negative polarity cases that was also seen in Fig.11(b)..
    (d) caption to Figure 12: its simply incorrect that the figure shows nega-
    tive (left column) and positive (right column)  this is only in panels (c,d).
    (e) p.17, line 5: should be Figure 12(b), not 11(b).
    (f) caption to Figure 14: perhaps rewrite wave of elevation and wave of

depression in terms of polarity (as elsewhere in text).
(a) **This has been changed.**
(b) **The sentence has been changed.**
(c) **The wording has been changed.**
(d) **This has been changed.**
(e) **This has been changed.**
(f) **The wording has been changed.**

28. Will the URLs be removed from the final reference list?
**Unfortunately, this appears to be due to the NPG bibliography style, note of this will be sent with submission.**

29. caption to Table 1: obvious typo with superscript.
**This has been fixed.**

**Reviewer 2**

**We appreciate the reviewer's time, and many of the suggestions provided are excellent. We certainly apologize for the transposition of the author list that led to the absence of the discussion of Lelong and Sundermeyer (JPO 2005), and have included extensive discussion of this paper in the revisions. We have focussed the revised manuscript on evidence of nonlinear effects, providing linear theory early on both in terms of links to mathematical physics (through the Klein Gordon equation) and the predictions of linear dispersive wave theory. We follow this up with a detailed comparison with Lelong and Sundermeyer (finding many points of agreement), though it should be noted that this is a secondary goal of our work. As per the reviewer's suggestion the Froude number is discussed, and since this is essentially an inviscid process the confusing discussion of the Reynolds number has been removed. We have confirmed that the simulations are Reynolds number independent. However, we do disagree with the reviewer rather fundamentally on the role of numerical simulations. Numerical simulations are not only useful when they are compared against model theories. This is certainly useful, and one of us has worked on aspects of this problem in the past. Good numerical simulations allow for a virtual laboratory in which 'what if' questions can be asked. Here resolution matters, and while we did not dwell on this in the manuscript, the numerical set up in the Lelong and Sundermeyer paper cannot answer questions about the radiating wavetrain in any situation apart from that of a very broad initial condition which yields only small amplitude, long waves (even here the periodic boundary conditions are difficult for us to interpret). Such a situation is precisely the opposite of what one finds in the laboratory experiments that provided the major motivation for our work. Perhaps our presentation led to the big picture being missed, and we hope the revisions highlight the novel aspects of what we have done.**

**We provide detailed responses below (though we note that much of the text has changed).**

The writing and presentation has improved on this resubmission but, unfortunately, the depth of analysis has not. Results of numerical simulations of mass adjustment for the classic dam-break problem with nonlinear high-resolution 2D numerical simulations are presented, with a focus on the dispersion characteristics of the radiating wavepacket, on the impact of the initial disturbance polarity and the role of the Rossby number. The authors are applied mathematicians, yet only 4 equations appear in the paper: the equations of motion and a KdV equation. In fact, it is not clear why the latter equation is even included since it is not solved nor used at all in the manuscript. Comparisons with previous work are concerned to very broad generalizations. Comparison of simulations in different regimes are equally qualitative. We are not told why Ro=1 represents a transition in behavior. What is the impact of the Reynolds number? Why not mention the Froude number? The analysis remains primarily concerned to looking at kinetic energy Hovmoller plots and isopycnal displacements. Numerical simulations can be a powerful tool when used in conjunction with some theory, but here no hint of theory is presented. What have the authors learned from this study? How does energy in the geostrophic state and radiating waves depend on the Rossby number or the nonlinearity parameter? What is the role of the Reynolds number? Why include an entire table of Reynolds numbers when the impact of dissipation is not ever discussed? In my first review, I mentioned that the authors might compare their results with Lelong and Sundermeyer (JPO 2005). Instead, they concentrated on Sundermeyer and Lelong (JPO 2005) which, I agree, has no bearing on the current study. I am sorry that I cannot recommend publication of this manuscript in its present incarnation.

**Many of the responses to the general comments can be found in the preamble. As a point of fact, the KdV equation was (and continues to be) utilized immediately after its definition to explain the importance of disturbance polarity in the type of response (solitary wave train versus undular bore) for the nonrotating case. While this is not 'solving the equation' it is, nevertheless, useful. The variations in the Rossby number are, in our opinion, much easier to understand in the revised version which provides detailed comparisons with the work of Lelong and Sundermeyer. In terms of what we have learned in the study, we are quite clear about the fact that the primary contribution is to catalogue the nonlinear effects both in the wave train and the geostrophic state. Detailed comparison to previous work centers on the papers by Kuo and Polvani and Lelong and Sundermeyer, with direct referencing of relevant figures in the latter. The vertical structure of the inertial oscillations is also considered something that is not possible to do using shallow water theory, or boundary conditions that are periodic in the vertical.**

1. Abstract, line 7: How do variations in the Rossby number demonstrate

the presence of two wave trains?

**The wording has been changed.**

2. Page 3, line 32: Since the set-up is on experimental scales, the at ocean bottom should be at bottom.

   **The sign has been changed.**

3. Page 4, As in the previous version, the definition of → on line 7 is still not consistent with the expression for → given on Page 6 (top of page).

   **This has been made consistent.**

4. Page 4, top of the page, equations 1-3: These are the Navier-Stokes equations, not the Euler equations which are, by definition, inviscid.

   **This has been corrected.**

5. Page 4, line 15: Why not state Lamb's equation 14 explicitly? Nowhere in the manuscript does an energy equation appear. Which terms are dominant in the different regimes?

   **This has been removed.**

6. Page 7, paragraph starting on line 5: It not clear why the paragraph on the KdV theory is included since no explicit comparison with numerical results is made. You are solving the NS equations, not the KdV or DJL equation. Do your numerical results match the solutions of either of these equations? In Eq (4), what is B?

   **The definition of $B$ has been provided. The reason for including the KdV equation has been discussed above. The DJL equation is discussed briefly to provide context. In the revised manuscript linear rotating theory is discussed as well, to provide a link with the well studied Klein Gordon equation.**

7. Page 8: Please state the polarity of your base case.

   **A statement has been added.**

8. Page 9, lines 10-11: Those 2 sentences can be combined: Figure 4(a) corresponds to the base case (Ro=1/2), 4(b) to $f = f_0/2$ (Ro=1) and 4(c) to $f = f_0/4$ (Ro=2).

   **The wording has been changed.**

9. Page 9, Sentence starting on line 19 does not make sense.

   **The wording has been changed.**

10. Page 9, lines 28-30 run-on sentence.

    **The sentence has been changed.**

11. Page 11, lines 8-9: '... since the potential energy may be zero since it can reach its initial starting point'. Do you mean to say that $\Delta$PE can be 0? What's wrong with that? The difficulty in computing $\Delta$PE/$\Delta$KE may come from $\Delta$KE vanishing at t = 0 but typically, one is interested in the

value of this ratio at large time.

**There was an error in the original ratio, it has since been fixed. The typical measure (seen in Kuo and Polvani 6.b) is $\Delta$KE/$\Delta$PE which is undefined at $\Delta$PE=0.**

---

## Author Response (AR3)

**Response to review**

We would like to thank the reviewer for spotting the various error in the text. All of the highlighted comments have been addressed as was recommended by the reviewer. Another re-reading has also been done to spot any more errors. We would like to comment that previous reviewers had mentioned the d.o.i. and website information in the references was distracting. However, the display of this information is controlled by the Copernicus Bibtex style which we have elected not to edit as per the manuscript preparation guidelines.